# Lamin A molecular compression and sliding as mechanisms behind nucleoskeleton elasticity

Alex A. Makarov [1], Juan Zou[1], Douglas R. Houston[1], Christos Spanos [1], Alexandra S. Solovyova[2], Cristina Cardenal-Peralta[1], Juri Rappsilber [1,3] & Eric C. Schirmer [1]

Lamin A is a nuclear intermediate filament protein critical for nuclear architecture and mechanics and mutated in a wide range of human diseases. Yet little is known about the molecular architecture of lamins and mechanisms of their assembly. Here we use SILAC cross-linking mass spectrometry to determine interactions within lamin dimers and between dimers in higher-order polymers. We find evidence for a compression mechanism where coiled coils in the lamin A rod can slide onto each other to contract rod length, likely driven by a wide range of electrostatic interactions with the flexible linkers between coiled coils. Similar interactions occur with unstructured regions flanking the rod domain during oligomeric assembly. Mutations linked to human disease block these interactions, suggesting that this spring-like contraction can explain in part the dynamic mechanical stretch and flexibility properties of the lamin polymer and other intermediate filament networks.

[1] Wellcome Centre for Cell Biology, University of Edinburgh, Max Born Crescent, Edinburgh EH9 3BF, UK. [2] Institute for Cell and Molecular Biosciences/NUPPA, The Medical School, University of Newcastle, Framlington Place, Newcastle upon Tyne NE2 4HH, UK. [3] Chair of Bioanalytics, Institute of Biotechnology, Technische Universität Berlin, Berlin 13355, Germany. Correspondence and requests for materials should be addressed to J.R. (email: juri.rappsilber@ed.ac.uk) or to E.C.S. (email: e.schirmer@ed.ac.uk)

Close to 300 mutations in a single nucleoskeletal protein of largely unknown structure—lamin A —are linked to 13 distinct human syndromes ranging from cardiomyopathy to lipodystrophy and progeria. One hypothesis to explain this wide range of pathologies from mutations in one protein is that mechanical weakening of the nucleoskeleton underlies disease[1]. This hypothesis is supported by observations that nucleoskeletal stiffness affects tissue differentiation, tissue maintenance, and metastatic invasion[2–4]. Yet knowledge of lamin molecular structure and filament assembly—or of other intermediate filaments—is largely limited to electron microscopy[5–7] and fragment crystal structures[8–11]. For example, for lamin A our insight is limited to crystal structures of the Ig domain representing 1/6th of the protein and two contradictory structures of smaller overlapping fragments at the end of its distended rod domain[8–11].

Intermediate filaments are the most tensile of the major cytoskeletal systems[12]. Unlike actin and tubulin, intermediate filaments are diverse both within humans who have 70 different intermediate filament genes and in evolution[13]. The main structural feature of intermediate filament proteins is their central distended α-helical rod domain, predicted to be made of 3–4 separate coiled coil segments that drive their dimerisation, and is flanked by usually less structured head and tail domains of variable length contributing to further polymerisation[14,15]. Lamins similarly have a short N-terminal region followed by a distended rod domain and a more variable globular C-terminal region. They are likely the progenitor intermediate filament[16,17] and differ from cytoplasmic intermediate filaments in having an NLS for nuclear targeting, a CaaX box for farnesylation, and additional α-helical sequences (6 heptads) inserted into the second coiled coil segment. A recent cryo-electron tomography study of the nucleoskeleton polymer from HeLa cells currently provides our most detailed to date visualisation of the assembled structure of lamins;[7] however, molecular details of lamin structure, assembly intermediates, and interactions within the polymer remain elusive. In fact, no molecular structure of any full-length intermediate filament protein has been solved due to the flexibility and poor solubility of their predicted central α-helical coiled coil rod domain[8,18,19].

Here we determine interactions in lamin dimers and tetramers supporting lamin assembly, finding evidence for the predicted parallel coiled coil structure and for intra-molecular compression and inter-molecular sliding supporting nucleoskeletal polymer stress-responses. Specifically, we develop a SILAC cross-linking mass spectrometry (CLMS) approach to distinguish intra- and inter-molecular interactions within lamin homomers: between lamin A molecules within dimers and between lamin dimers at tetrameric assembly stages in solution. This reveals that three flexible linker regions in the lamin A coiled coil rod can electrostatically drive sliding of adjacent coiled coil segments onto each other to achieve a spring-like contraction in rod length. We further show how both head and tail unstructured regions flanking the rod can act as dynamic polar bridges stabilising the lamin A tetrameric interface and likely defining their assembly dynamics[20,21]. Importantly, several tested disease-causing mutations disrupt these properties. Our results suggest an alternative mechanism where unstructured head, tail and linker regions allow reversible small-scale deformation of the nucleoskeletal filaments without the drastic α-helix unfolding of the more stable coiled coil segments postulated to govern intermediate filament mechanics[22]. This model explains variations in appearance of nucleoskeletal filaments[7] and provides a molecular explanation for many uncharacterised disease mutations in lamin A.

## Results

**Lamin rod length varies from 40 to 52 nm.** A recent cryo-electron tomography study of the intact nucleoskeleton polymer from HeLa cells[7] suggested a reduction in rod length from the historically measured ~ 52 nm[5,23] to ~ 40 nm based on decreased spacing between globular C-terminal densities in the assembled filaments. The difference between in vivo and in vitro measured rod lengths could reflect a change in rod length upon assembly and/or differences in the buffer conditions used for in vitro studies. The mechanism of rod shortening in either case could in theory derive from dimer sliding resulting in increased rod overlap (2–3 nm)[24] or from rod shortening by ~ 15–20% (Fig. 1a) or from both.

In search of support for the shortening hypothesis we thought to re-analyse distribution of lamin A rod length in conditions favouring lamin dimerisation[5,23,25] by means of rotary metal shadowing EM[26]. To tackle observed differences in rod lengths in vitro and in vivo recombinant lamin A dimers purified from bacteria[27] were equilibrated in either the conventional lamin in vitro dimerization Tris buffer (25 mM Tris, 250 mM NaCl, pH 8.0) or in higher ionic strength sodium phosphate (NaPi) buffer (100 mM NaPi, 250 NaCl, pH 8.0)—similar to the one used for visualisation of the intact nucleoskeletal polymer[7]. Measuring 300 dimers visualised by rotary metal shadowing EM revealed rod lengths ranging respectively from 40.7 to 56.9 nm and from 41.0 to 56.0 for 90% of measurements in Tris and NaPi buffer, with a significant trend towards shorter rods in NaPi buffer: median rod length of 51.1 nm in Tris buffer vs 49.2 nm in NaPi buffer (p-value $4.45 \times 10^{-3}$) (Fig. 1b, c and Source Data file). This also indicates that buffer conditions can contribute to the rod shortening. A similar shortening was indicated by analytical ultracentrifugation where an increase in more compact species with S-values higher than the 3.81 S calculated for a dimer with a 51 nm long rod was observed in NaPi buffer (Fig. 1d). These species are likely also dimeric because, while lamin A head-to-tail tetramers can be clearly observed by EM in either buffer, these are exceedingly rare (~ 1 per 20–25 dimers in NaPi) (Supplementary Fig. 1a and Source Data file); and because a separate population of species with S-value roughly fitting the calculated 4.9–5.0 S for lamin head-to-tail tetramers is readily detectable in analytical ultracentrifugation experiments (Supplementary Fig. 1b). This evidence suggests an existence of compressible elements within the lamin rod.

**Cross-linking lamin dimeric and tetrameric assembly states.** The observed spread of lamin A rod lengths is a further reflection of its known rod flexibility. Unfortunately there is currently no methodology in place for isolation of dimers in each particular compression state to investigate interactions contributing to these states. Therefore, to better define lamin A structure and investigate whether compression derives from dimer sliding or rod shortening we turned to CLMS. This method is capable of capturing and reporting residue interactions across a population of flexible protein molecules in different structural states as opposed to relying on all protein molecules being in exactly the same state. EDC (1-ethyl-3-(3-dimethylaminopropyl)carbodiimide hydrochloride) was chosen as the cross-linking reagent for this study as one without a linker arm (zero-length cross-linker): it requires respective side chains to be in immediate proximity providing greater cross-linking resolution. It is also heterobifunctional and thus captures actual electrostatic and/or polar interactions within the protein molecule: specifically between side chain carboxyl groups of aspartic and glutamic acid residues and primary amine groups of lysine (and N-terminal methionine) or hydroxyl side chain groups of serine, threonine and tyrosine (residue pairs E/D-K/S/Y/T).

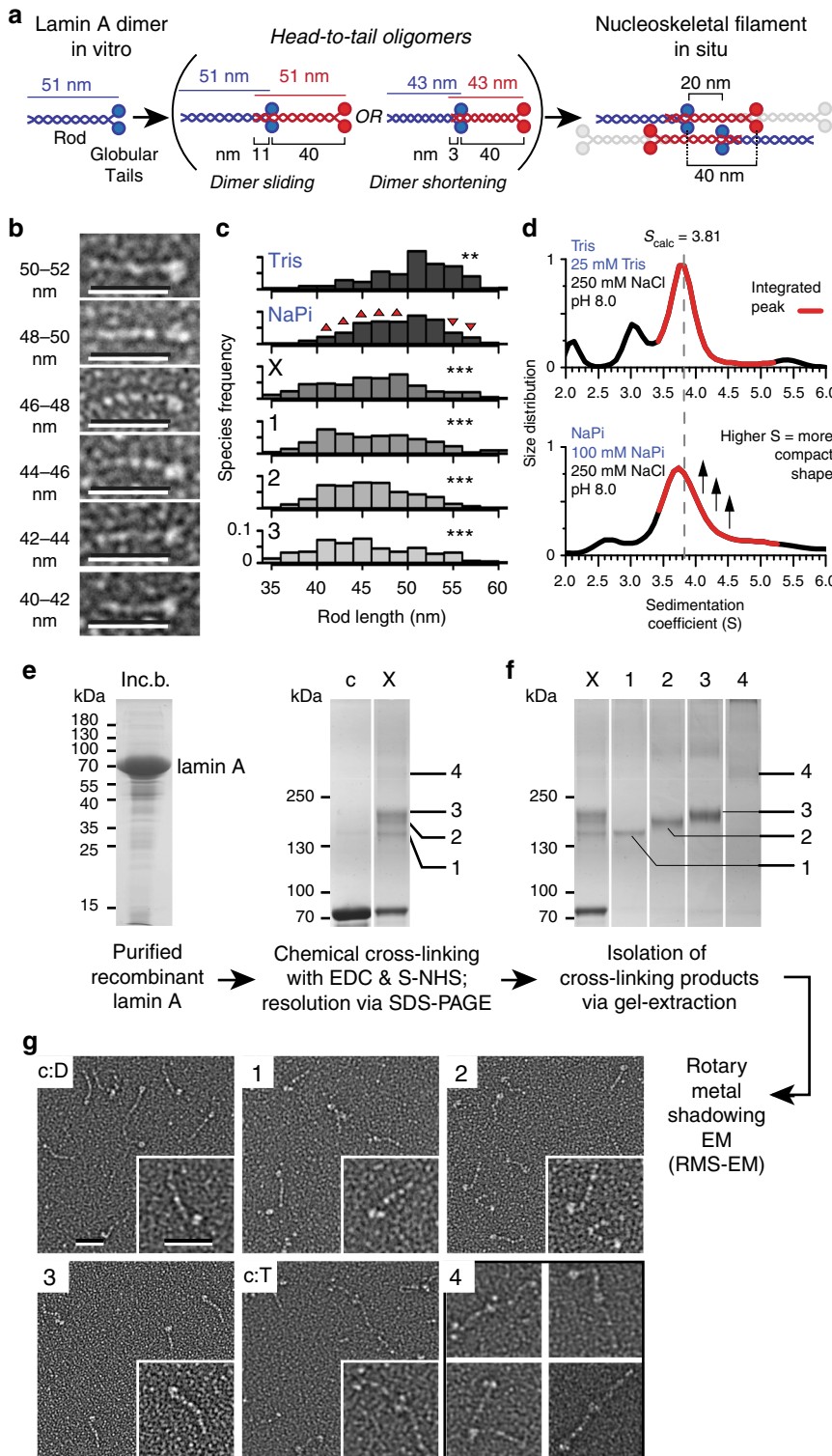

Chemical cross-linking of recombinant lamin A with a mixture of EDC and S-NHS (N-hydroxysulfosuccinimide—a stabilising helper agent) appeared capable of capturing lamin A dimeric structure in its various states of compression as well as its ordered head-to-tail assembly. Cross-linking was carried out in the NaPi buffer in which we noted the significant variation in the dimer rod length (Fig. 1c). Cross-linking yielded three major product-bands (Fig. 1e, bands 1–3) and several minor products (bands 4 and above), which upon isolation by gel-extraction and visualisation with rotary metal shadowing EM (Fig. 1f, g) were

revealed to contain respectively mostly dimeric (1–3) and tetrameric species (4). The latter was expected because head-to-tail tetrameric species are readily observed alongside dimers in the un-cross-linked control in NaPi buffer (c:T, c:D and Supplementary Fig. 1a). Note that isolated cross-linked species were similar in appearance to those in the un-cross-linked control indicating that cross-linking preserved original lamin architecture. Rod measurements of the dimer-rich cross-linked material further revealed that cross-linked dimers, while widely distributed for lengths, were significantly shorter than those in the

**Fig. 1** Chemical cross-linking captures variation in lamin A rod length. **a** Rod shortening (middle-left) or increased head-to-tail overlap (middle-right) may explain nucleoskeletal filament characteristics observed in situ. **b** Example lamin A dimers with rods varying from 40 to 52 nm visualised by rotary metal shadowing EM. Scale bars, 50 nm. **c** Frequency distribution of lamin A rod length measured from EM on uncross-linked lamin A in Tris and NaPi buffers, whole cross-linking reaction (X) and individual cross-linked bands in NaPi buffer (1–3) (see **f**). Bars, 2 nm; n = 300 dimers per condition; p-values (Kruskal-Wallis and Dunn with Holm correction) for differences from NaPi: Tris $4.45 \times 10^{-3}$ (**), X $9.85 \times 10^{-11}$ (***), band-1 $2.43 \times 10^{-7}$ (***), band-2 $9.52 \times 10^{-11}$ (***), band-3 $2.51 \times 10^{-15}$ (***). **d** Analytical ultracentrifugation particle size-distributions of lamin A in Tris and NaPi buffers reveals increased frequency of more compact higher S-value species (arrows). **e** Lamins purified from inclusion bodies (left panel, inc. b.) were chemically cross-linked with a mixture of EDC and S-NHS (middle panel), yielding four major oligomeric products (1–4). **f** These bands were successfully purified by gel-extraction as evidenced by SDS-PAGE (right panel) of lamin A before (control, c) and after cross-linking with EDC:S-NHS (X) and individual gel-extracted cross-linked bands (1–4). **g** Rotary metal shadowing EM of cross-linked protein in bands 1–3 reveals principally dimers (control c:D) while band 4 reveals tetramers (control c:T). Source data for **c**, **e** and **f** are provided in the Source Data file

uncross-linked NaPi control (Fig. 1c, 1–3), suggesting that interactions leading to rod compression must have been captured by chemical cross-linking and should be identifiable by subsequent mass spectrometry analysis. Similarly cross-linked lamin head-to-tail tetrameric species should, in theory, provide information about the degree of rod overlap and concurrent interactions at the head-to-tail interface.

**SILAC chemical cross-linking**. To distinguish interactions supporting lamin A homo-dimer subunit assembly and mechanics from those supporting its homo-polymer assembly (i.e. head-to-tail tetramers), we further developed an original CLMS approach utilising EDC chemical cross-linking and SILAC labelling. Cross-linking of a mix of light and SILAC-labelled heavy lamin A Homo-Iso-Dimers (Fig. 2a, HIDm) enabled distinguishing intra-from inter-dimeric interactions. Cross-linking of a mix of Homo-/hetero-Iso-Dimers similarly enabled distinguishing intra-and inter-molecular interactions within dimers (Fig. 2a, H/hIDm). Lamin A dimers do not exchange chains even in denaturing conditions of 6 M urea as is indicated by a separate elution of un-tagged and 6xHis-tagged lamin A dimers in Ni-NTA pull-down experiments (Supplementary Fig. 2). Thus to obtain the homo-/hetero-iso-dimer mix we artificially drove chain exchange by refolding gel-extracted monomers, as has been done previously for other intermediate filaments[28,29]. As a proof-of-concept this was again initially attempted on a 1:1 mix of un-tagged and 6xHis-tagged lamin A monomers and formation of hetero-dimers was confirmed in Ni-NTA pull-down experiments showing clear retention of ~ 50% of un-tagged lamin A on the beads and its co-elution with all of the 6xHis-tagged lamin A (Supplementary Fig. 2a). Co-gel-extraction and refolding of light and heavy lamin A monomers thus results in a homo-/hetero-iso-dimer mix comprised of 50% of light-heavy and 25% each of light-light and heavy-heavy dimers (Fig. 2a). An additional in vitro assembly assay was employed to confirm normal structure and assembly-competence of gel-extracted and refolded lamin A dimers (Fig. 2b).

Chemical cross-linking of both homo-iso-dimer mix and homo-/hetero-iso-dimer mix with EDC/S-NHS yielded identical banding patterns of cross-linked products (Fig. 2c). This allowed a parallel analysis of identical residue interactions in both lamin A samples and consequent identification of their intra-/inter-molecular origin: to this end we adapted the comparative cross-linking analysis routine[30,31] to record the frequency with which cross-links between these residues occurred between pairs of similarly and differently labelled peptides (Fig. 3, "Methods"). In the cross-linking experiment with homo-iso-dimer mix (HIDm X) a residue interaction within a single dimer should always be represented by cross-links between either two light peptides (LL) or two heavy peptides (HH), while an inter-dimeric residue interaction would be represented by cross-links between all 4 combinations of peptides (LL, HL, LH and HH) (Fig. 3b).

Similarly in the homo-/hetero-iso-dimer mix cross-linking experiment (H/hIDm X), a cross-link occurring only between similarly labelled pairs of peptides would be a cross-link happening strictly within a single molecule of lamin A (Fig. 3c) and thus captures an intra-molecular residue interaction. Cross-links occurring between all possible combinations of peptides would be inter-molecular and occurred between proximal residues either on two opposite chains of a single dimer, or on separate chains of two interacting dimers. Thus, for each cross-link the frequency of occurrence between LL, HL, LH and HH peptides was recorded as a function of peptide ion intensities in MS1. This was done manually in Xcalibur (Thermo) for all cross-links or—for a subset of cross-links—semi-automatically using Skyline[32] across respective extracted ion chromatograms (XiC) (Supplementary Data 1–3) and corresponding frequencies of this particular cross-link occurrence as inter-dimeric and inter-molecular were further determined (Fig. 3d). Such analysis of cross-links between the same pair of residues in HIDm X and H/hIDm X experiments done in parallel thus enables further calculation of the frequency with which these two residues interacted/were proximal as intra-chain or inter-chain within the dimer and between two dimers (Fig. 4 and Supplementary Data 4). A total of 233 unique cross-linked residue pairs (subsequently referred to as just cross-links) from dimer-rich bands 1–3 and 143 cross-links from tetrameric band 4 were identified. Fittingly, SILAC quantification revealed that 229 out of these 233 unique cross-links from dimer-rich bands indeed occurred in dimers with varying frequency while 66 out of 143 cross-links from the tetrameric band were inter-dimeric (Fig. 4).

**Structural compression mechanism in the lamin A rod**. Intermediate filament rods are thought to consist of multiple α-helical segments—coils 1A, 1B and 2—that dimerise into parallel heptad or hendecad coiled coils[33–36] accounting for ~ 48 nm of rod length by prediction. A heptad is a 7 amino-acid motif that drives dimerisation of two alpha helices into a stable twisted superhelix compared to a hendecad which is an 11 amino acid motif that forms a less stable parallel bundle of alpha helices. Most of the lamin A coiled coil rod is predicted to feature heptad structure, but at the end of coil 1B and beginning of coil 2 a hendecad structure was predicted to form[34,37] (Fig. 5a). Individual coils are connected by linkers (L)—both α-helical and unstructured—thought to contribute 3.4 nm[38], bringing the rod to ~ 51 nm (Fig. 5a). A total of 126 cross-links were identified between pairs of residues within the rod and an additional 104—with at least one residue of the pair outside the rod—the head or the tail domains (Fig. 5b, c and Supplementary Data 5, 6). Cross-links within the rod should be informative of its structure, its potential flexible regions and thus of interactions leading to the observed rod shortening. These cross-links were therefore checked against the predicted rod structure starting with those within individual coiled coil segments (see "Methods", Supplementary Discussion).

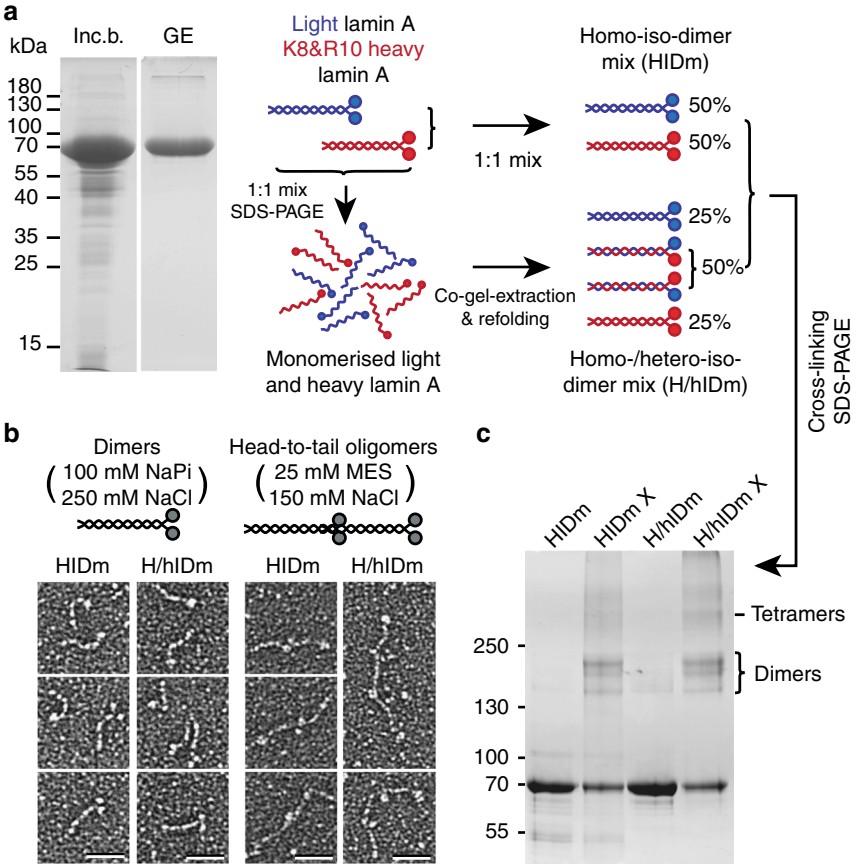

**Fig. 2** Lamin A gel-extraction and refolding prior to SILAC CLMS. **a** Bacterially expressed light (blue) or isotopically-labelled heavy (red) lamins were purified from inclusion bodies (inc. b.). To determine interactions at the inter-dimeric interface a 1:1 light:heavy Homo-Iso-Dimer mix (HIDm) was analysed. To distinguish intra- vs inter-molecular cross-links, the light and heavy lamins were mixed 1:1, gel-extracted (GE), and refolded, generating a Homo-/hetero-Iso-Dimer mix (H/hIDm) of 1:1:2 light homo-iso-dimers, heavy homo-iso-dimers and light/heavy hetero-iso-dimers mix for cross-linking. **b** Rotary metal shadowing EM indicates that the gel-extracted and refolded H/hIDm lamin A retains structure similar to native HIDm lamin A (micrographs, left column) and both are assembly competent (right row). Scale bars, 50 nm. **c** Coomassie staining of material before and after cross-linking (X) showing identical cross-linking product patterns for HIDm and H/hIDm lamin A. Source data for **a** and **c** are provided in the Source Data file

A total of 50 such cross-links were identified within the coiled coil segments, 31 of which supported the predicted coiled coils in parallel dimers with some anticipated irregularities in coil segment termini packing: specifically low stability of the coil 1A N-terminus[39] and PH region[40], a stutter after L1[38,41] and hendecad-heptad transitions in the coil 1B C-terminus and after the PH[35–37] (Fig. 6a blue and orange cross-links, Supplementary Fig. 3, Supplementary Data 5 and Supplementary Discussion for irregularities). Importantly, as all of these cross-links either already fit the predicted linear coiled coil structure or can be satisfied via the axial chain rotation and small interruptions in the individual chain α-helical structure to bring respective residue side chains in contact for EDC cross-linking, none of these cross-links can directly account for rod shortening. Interestingly, of the remaining 19 cross-links within individual coiled coil segments, 16 cross-links were between residues in the coiled coil segment termini but too distal for EDC cross-linking in the predicted coiled coil structure even with small structural changes; thus they imply more severe irregularities in these termini (Fig. 6a, red cross-links). Three similar cross-links were also found between residues within in the middle of the coil 1B. However all of these cross-links could together account for less than half of the observed rod shortening, suggesting that rod shortening cannot be due to just the captured irregularities in coiled coil segment packing.

By contrast and more excitingly, many cross-links were identified between adjacent coiled coils or between linker regions and coiled coils. A total of 76 such cross-links were found at the linker regions L1, L12 and two additional putative linker regions: the once thought classical linker (L2) in the second hendecad of PH region[38] and the linker (L3) at the hendecad-heptad transition after the PH[36], also previously predicted in[42] (Fig. 6b, c, Supplementary Discussion). While flexible linker regions could easily fold back onto adjacent coiled coils to satisfy some (though not all) of the linker-to-coil cross-links, most coil-to-coil cross-links would be too distal for EDC to accommodate in a fully linear rod (Fig. 7a, linker exclusion). Instead these cross-links would require adjacent coiled coil segments to overlap in a relatively parallel tandem stagger or in an anti-parallel fold (Fig. 7a, bottom schematics). Based on the maximal linker extension a group of 55 cross-links tightly clustered in and around the linker regions (Fig. 6b, Supplementary Data 5) were estimated as being possible to satisfy without the rod folding over in an anti-parallel fashion. We thus propose that the observed linear rod shortening may be due to flexible linkers allowing sliding of adjacent coiled coil termini onto one another in a tandem stagger.

To test this possibility, molecular docking in Rosetta with cross-linking constraints (cross-link guided docking)[43] was employed to search for potential coil termini overlap interfaces

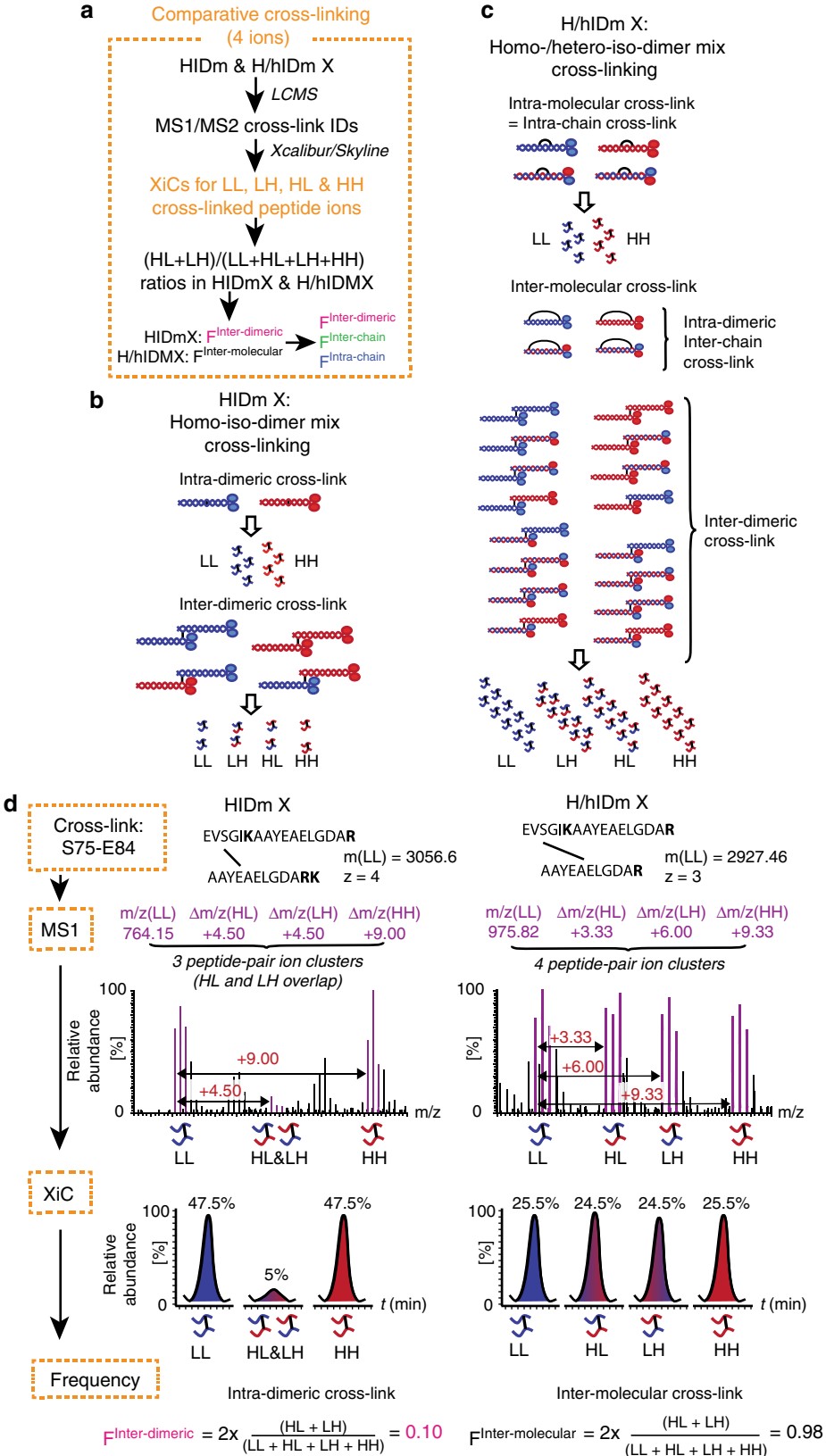

that are relatively stable and would satisfy coil-to-coil cross-links (28 out of 55). The main assumption used was that interacting coiled coil termini retain their α-helical geometry while linker regions are flexible. 100,000 structural models featuring docked adjacent dimeric termini fragments for coils 1 A and 1B, coil 1B

and PH, two halves of the PH, and PH and coil 2 were generated to satisfy sets of cross-links around respective linker regions L1, L12, (L2) and (L3); and Xwalk[44] was further used to search for models with stable interfaces that satisfied input cross-links (Fig. 7b, see "Methods"). Among these we found both the tandem

**Fig. 3** SILAC CLMS distinguishes inter-dimeric and inter-molecular residue interactions. **a** The 4-ion comparative cross-linking analysis routine. **b**, **c** Mass-spectrometry of cross-linked and digested HIDm or H/hIDm allows distinguishing pairs of cross-linked peptides/residues originating from respectively intra-/inter-dimeric or intra-/inter-molecular cross-links. Note that inter-dimeric and intra-dimeric-inter-chain cross-links yield similar types of peptide pairs. **d** Principle behind the calculation of frequencies of cross-links occurrence between dimers $F^{Inter-dimeric}$ or between molecules $F^{Inter-molecular}$: for each cross-linked pair of peptides up to 4 precursor ion clusters are found in MS1 of each spectra, each with an $m/z$ fitting different degrees of labelling within the cross-linked peptide pair; corresponding extracted ion chromatograms (XiCs) are then determined for each of the mono-, first and second isotopic peak in each cluster—manually in Xcalibur or semi-automatically in Skyline; quantification of peak intensities across XiC areas ions yields information about intra-/inter-dimer/-molecular origin for each cross-link

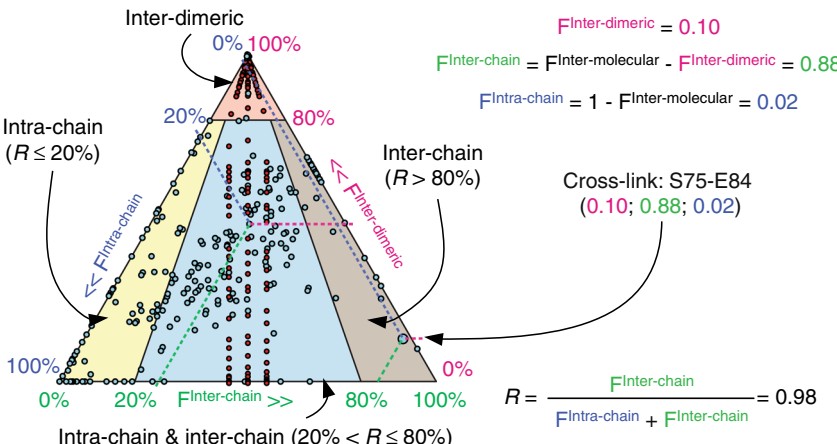

**Fig. 4** Mapping lamin A inter-dimeric, inter-chain and intra-chain residue interactions. Frequencies of occurrence as inter-dimeric $F^{Inter-dimeric}$ (magenta axis), intra-dimeric inter-chain $F^{Inter-chain}$ (green axis) or intra-chain $F^{Intra-chain}$ (blue axis) calculated as shown (see also Supplementary Data 4) and plotted on a triangle plot for each cross-link in bands 1–3 and band 4 (see also Supplementary Table 1). Band 4 was analysed only in the HIDm experiment, therefore for only the $F^{Inter-dimeric}$ for these cross-links. Cross-links were deemed as predominantly ($F > 80\%$) inter-dimeric (red sector), as intra-dimeric&intra-chain if $R \leq 20\%$ (yellow sector), intra-dimeric&inter-chain ($R > 80\%$, beige sector), intra-dimeric and both intra-&inter-chain ($20\% < R \leq 80\%$, light-blue sector)

stagger models where the linker engages interactions to fold the beginning of the linearly following coiled coil back over the end of the preceding coiled coil and anti-parallel models where the two coiled coils fold back onto one another making roughly a U shape. However, anti-parallel folding models were discounted because these are not consistent with the EM data that shows linear as opposed to folded back rods. Two anti-parallel folds in tandem would maintain the overall rod linearity, but only models with double folds in two of the three—L12, (L2) or (L3)—would not shorten the rod further than any of our measurements. Nonetheless, such possibilities cannot be completely ruled out in the context of an in vitro cross-linking experiment. The wide spread of the measured rod lengths without cross-linking and, more importantly, the downward shift of rod lengths upon cross-linking suggest a temporal nature of any compressed state. Therefore, a population of single-folded intermediate dimers should exist at any given time, which again contradicts collected EM data.

Particular cross-linked residues suggest that both tandem stagger and anti-parallel folding would occur via multiple electrostatic interactions (Supplementary Figs. 4 and 5). The majority of cross-links across L1 and L3 supported tandem stagger rod shortening by up to 5 nm each with minimal or no rod bending: four out of six coil-to-coil cross-links in both intra- and inter-chain variations and 1 only as intra-chain around L1; 11 out of 12 coil-to-coil cross-links around L3 in both intra- and inter-chain variations (see below and the "Methods" section). Some cross-links could obviously not co-exist with others within a single tandem stagger model (Supplementary Fig. 4a), but > 250 tandem stagger models with stable inter-coiled coil interfaces satisfying overlapping subsets of cross-links were found as

determined by Rosetta I_sc values < −5.0 (Rosetta's **I**nterface **sc**ore calculated as a difference between the total energy of the complex and the sum of total energies of each partner in isolation; as detailed in RosettaDock application documentation https://www.rosettacommons.org/docs) (Supplementary Fig. 4b, 5d and Supplementary Data 7–11). Cross-links also supported rod shortening for L12 with 50% supporting stable tandem stagger folding (3 out of 7, but 10 out of 20 intra- and inter-chain variations in total) and rod shortening by 3.5 to 4 nm. L2 cross-links yielded one stable model satisfying only one out of four cross-links and only in the intra-chain variation, arguing against a tandem stagger between the two halves of the PH region. Together this data clearly shows the possibility of stable coiled coil segment termini tandem stagger interactions that can satisfy observed cross-links and can account for up to ~ 10 nm of rod shortening (Fig. 7c).

Such rod shortening is also supported by data from cross-linking-free analytical ultracentrifugation experiments: the integrated peak of sedimentation coefficients (3.4–5.2S) for un-cross-linked lamin A dimers (Fig. 1d) encompasses calculated sedimentation coefficients ($S_{cal}$) for lamin A dimers in various stages of compression calculated using generated tandem stagger models (Supplementary Fig. 6). A multitude of obtained tandem stagger models, especially for adjacent termini of coils 1A and 1B and for PH C-terminus and the following part of coil 2, readily point to a high degree of redundancy in any rod shortening mechanism. The true extent of this redundancy became further apparent in an additional in silico docking experiment attempted for adjacent termini of coils 1A and coil 1B using Rosetta in an unconstrained mode: without cross-linking constraints to guide the docking. This generated a much larger set of relatively parallel

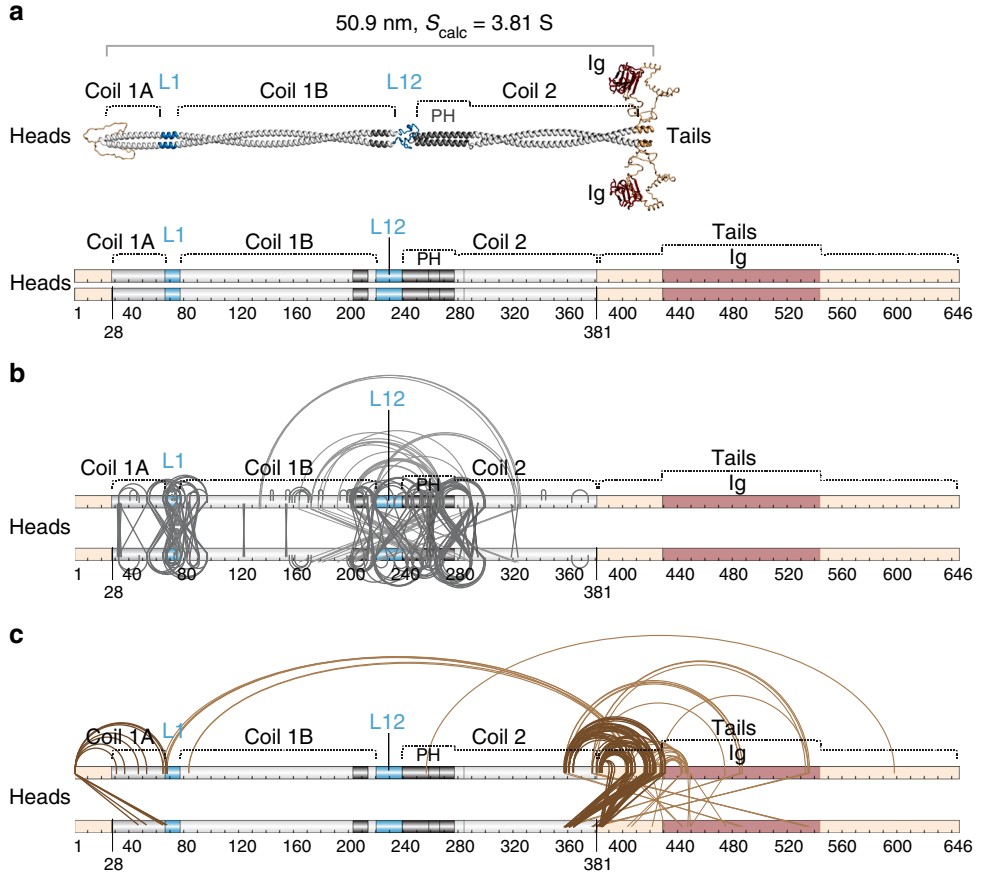

**Fig. 5** Distribution of intra-dimeric cross-links between lamin A dimer domains. **a** Lamin A schematics showing predicted structural organisation (top) and linear domains (bottom). Labels indicate predicted coiled coils (1A, 1B and 2 with PH parallel hendecad region; heptads light grey and hendecads dark grey) and linkers (blue–α-helical L1, unstructured L12). **b, c** Positions of all unique intra-dimeric cross-links identified (Supplementary Table 1) within the rod domain (**b**) or within the head and tail domains and between these domains and the rod domain (**c**). Cross-links within the rod appear to cluster around linker regions

stable tandem stagger models, of which almost 800 satisfied one or more of each of the 6 input cross-links in any of their intra- and inter-chain variations even when validated with Xwalk set to exclude models requiring additional rotamer manipulation for side-chains of cross-linked peptides (Supplementary Data 12). Similar modelling for dimeric coiled-coil fragments separated by L12 and L3 yielded 506 and 1071 stable parallel tandem stagger models respectively, again satisfying all of intra- and inter-chain variations for all but one of the input cross-links; but only 42 such models for PH fragments separated by L2 were found (Supplementary Data 13–15). Together this data provides a potential molecular explanation for observed rod compression/ bending where overlap of individual coiled coil termini is driven by the flexible linkers.

**Head and tail domain involvement in lamin compression**. Unstructured/flexible regions in the short N-terminal head and globular C-terminal tail domains also flank the rod; so we hypothesised they could similarly fold back over the coil termini to support the head-to-tail inter-dimer interface and sliding. Many intermediate filament head domains are positively charged: the vimentin head folds onto the negatively charged coil 1A[14,45] and is thought to facilitate dimer oligomerisation, presumably via interactions with coil 2[14,46]. In lamins both the head and the unstructured part of the tail flanking the rod are positively charged and crucial for head-to-tail oligomerisation[21,23]. Dimer

cross-links identified indicate that unstructured regions of the head and tail domains flanking the rod fold onto the adjacent rod termini (Fig. 8a–c). The head N-terminal methionine yielded seven cross-links to multiple coil 1A residues that support electrostatic interactions with its primary amine. The rod C-terminus parallels this with two negatively-charged sites in coil 2 yielding 56 cross-links to three positively-charged sites in the flanking unstructured tail region. Paralleling the rod linkers, these charged interactions can support multiple similar conformations to satisfy the observed cross-links (Supplementary Fig. 7 and Supplementary Data 6).

These head and tail interactions with the rod appear to additionally support head-to-tail assembly. Tetramer-band cross-links connected the head domain N-terminal methionine also to the coil 2 rod C-terminus. Furthermore, the positively-charged sites in the tail that in the dimer cross-linked band were only involved in shortening the rod, in the tetramer band now cross-linked also to negatively-charged coil 1A, L1 and coil 1B regions (Fig. 8d). The switch in interacting residues raises the possibility that the solitary dimer folding primes lamins for a strand exchange that stabilises tetramers (Fig. 8e, Supplementary Fig. 8 and Supplementary Data 6), though some dimer and tetramer cross-links could occur simultaneously (strand coordination). Different cross-links support these models with the two dimers overlapping by 3 heptads[24] or by all of coil 1A (Fig. 8f, g, Supplementary Fig. 9). Charged clusters supporting electrostatic interactions in both models thus could enable rod sliding in the

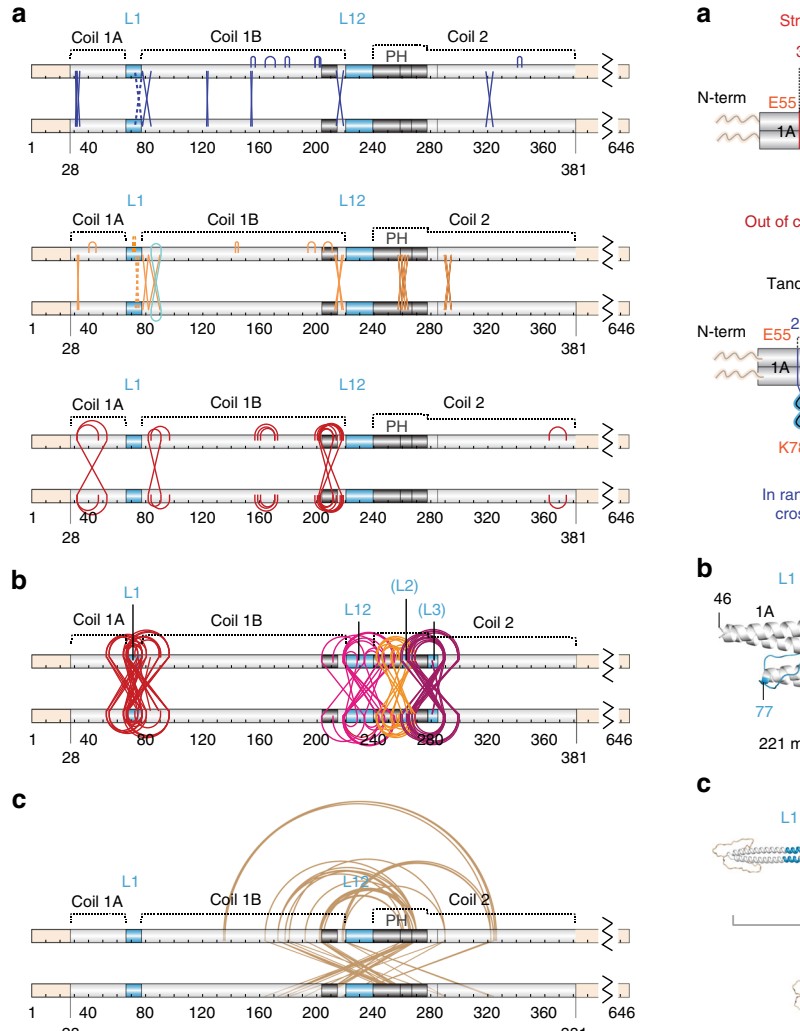

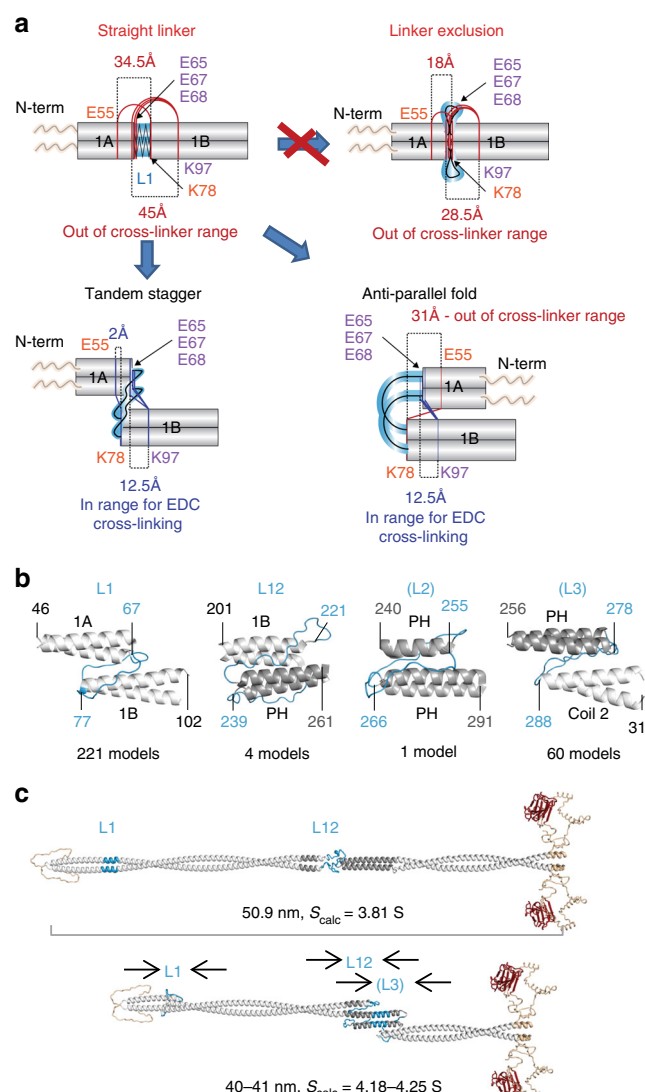

**Fig. 6** Categorisation of cross-links in the lamin A rod domain. Cross-links are indicated by the lines: arced lines above indicate intra-chain cross-links; crossed lines those between—inter-chain cross-links. **a** Lamin A schematics showing sub-categories of cross-links. A total of 31 cross-links were found in the lamin A dimer rod domain that satisfy classical coiled coil geometry with irregularities in the coiled coil termini and PH packng: 11 intra-chain, 19 inter-chain cross-links (blue and orange lines both) and 1 cross-link occurring both ways (cyan lines). Top: cross-links satisfying the physical constraints of the EDC cross-linker and supporting predicted heptad and parallel hendecad coiled coil structure of the predicted 51 nm rod. Middle: cross-links that can readily satisfy the cross-linker physical constraints in a parallel dimer with the indicated coiled coil segments, but indicate deviations from the predicted coiled coil structure: axial chain rotation, stutters and hendecad-heptad transition associated irregularities. Bottom: red lines indicate cross-links between residues with side chain at distances exceeding EDC constraints in the predicted structural model. Only such cross-links within the coiled coil regions are shown. **b** Cross-links around linker regions. Most identified cross-links clustered around L1 (dark red), L12 (light red) and two putative linkers L2 (orange) and L3 (brown) historically annotated within and after PH[36,42] (see Supplementary Discussion). **c** Gold lines indicate all remaining cross-links within the rod

**Fig. 7** Modelling of cross-links across linker regions can explain rod shortening. **a** Principal example models explaining rod shortening around linker L1. Residues in cross-linked pairs E55-K78, E65-K97 (both coil-1A-to-coil-1B) and E67/68-K97 (both L1-to-coil-1B) are too far away from each other (34.5, 48, 45 and 43.5 Å respectively) if L1 is assumed to be α-helical (Straight linker model). Complete exclusion of linker L1 would shorten the rod but is insufficient for bringing cross-linked residues within range for EDC cross-linking (Linker exclusion model). In contrast, tandem stagger folding of the ends of coiled coils over one another or an anti-parallel folding in a U-shape can accommodate the cross-linker constraints and so could explain rod shortening. The tandem stagger model implies a role of the linker in sliding the coiled coil segments onto each other. **b** Example tandem stagger models with stable coiled coil termini interfaces (I_sc < −5.0) generated via cross-link-driven docking in Rosetta with Xwalk validation and linkers reconstructed with SWISS-MODEL or MODELLER that support the linear rod shortening via coil sliding. **c** Model of distended linear lamin A dimer and an example model of dimer compressed via three sequential tandem staggers. When the tandem stagger is applied to three linkers in the rod it can shorten the rod from 50.9 nm to 40–41 nm, consistent with measurements obtained from analytical ultracentrifugation. $S_{calc}$ determined using SoMo (Supplementary Fig. 6 and Source Data file)

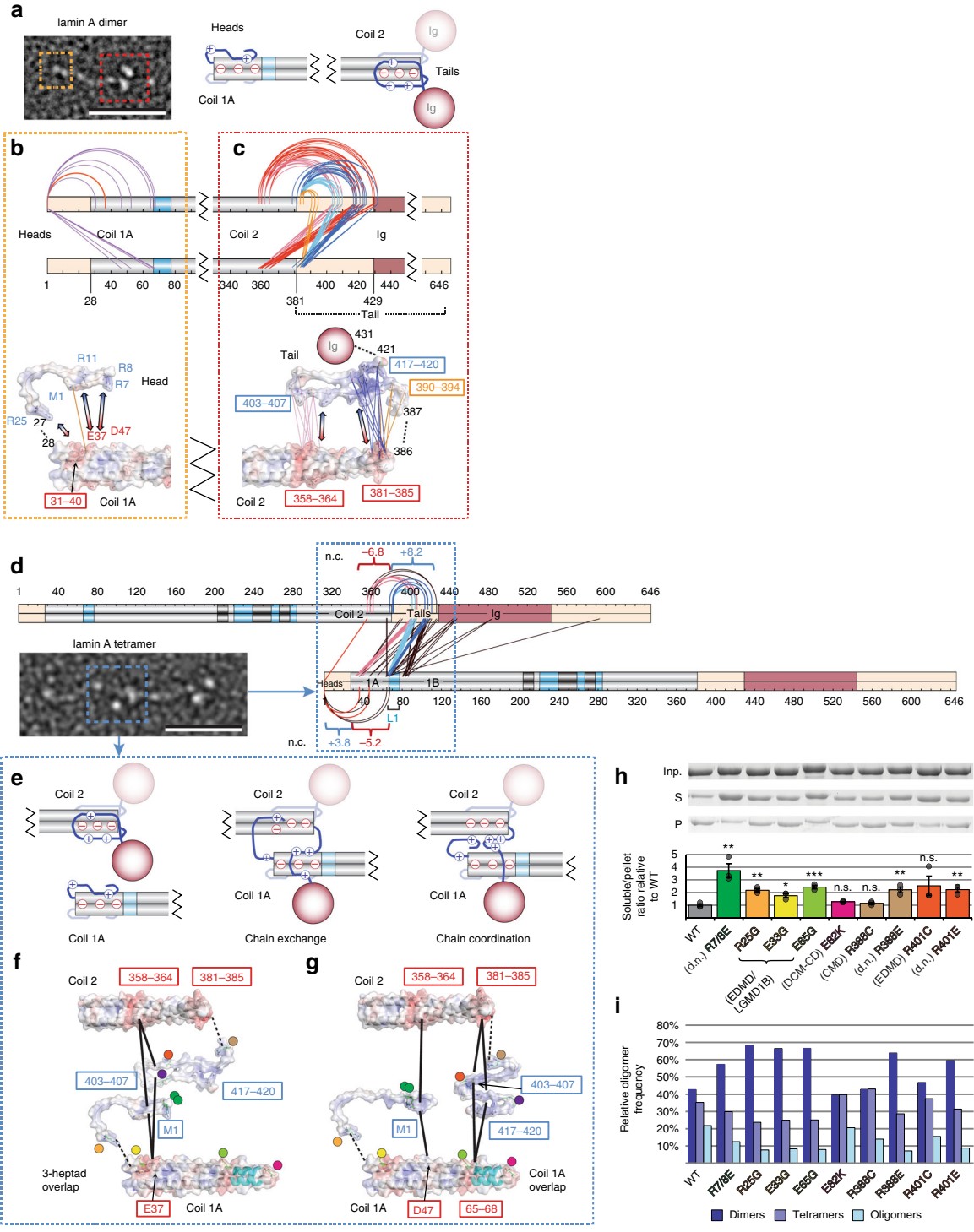

head-to-tail interface as an additional tensile mechanism, consistent with variable distances of C-terminal tail densities in assembled filaments in situ[7].

**Disease mutations disrupt the head-to-tail interface.** The proposed electrostatically-driven head-to-tail interface encompasses several residues mutated in patients with Dilated Cardiomyopathy (DCM-CD) or Emery-Dreifuss, Limb-Girdle and Congenital Muscular Dystrophies (EDMD, LGMD1B, CMD). Thus, we tested whether these mutations disrupt the interface using solubility and assembly assays. Substituting rod-flanking region arginines with disease-causing mutations or residues reversing charge (Fig. 8f, g coloured dots) increased solubility in conditions promoting lamin A polymerisation (Fig. 8h). Substituting glutamic acids with glycines in coil 1A achieved similar effects, while such substitutions further away in coil 1B had no effect. Direct examination of assembly stages by rotary metal shadowing EM on >1000 particles for each mutant revealed, compared to wild-type, an increase in free lamin A dimers relative to head-to-tail oligomers (Fig. 8i). Thus many disease mutations likely inhibit assembly by disrupting head-to-rod-to-tail interactions.

**Fig. 8** Electrostatic interactions drive head-to-tail docking in lamin A tetramers. **a** Head and tail domains fold onto their respective rod termini in solitary dimers. **b**, **c** This is supported by multiple head-coil 1A (**b**) and tail-coil 2 cross-links (**c**). Tail-coil 2 cross-links are coloured according to cross-linked sites (boxed). The head-coil 1 A cross-link (orange) and two sets of tail-coil 2 cross-links (pink and dark blue) were used to build the example models at the bottom that show head and tail regions, when aligned against adjacent rod termini, can satisfy these sets of cross-links. Electrostatic surface potential reconstructions (red-to-blue colouring– negative-to-positive charges: −5 to +5 kT) indicate charged interactions (arrows) drive the folding. **d** SILAC-determined inter-dimer crosslinks between the head-coil 1A-L1–1B region and tail-coil 2 region of adjacent dimers in tetramers. Net electrostatic charges (n. c.) of interacting regions are indicated. **e** Two proposed modes of lamin A head-to-tail tetramerisation: head and tail strands are exchanged between adjacent dimers termini or interact/coordinate with both. **f**, **g** Example models of the lamin A tetrameric interface with 3-heptad or full coil 1A overlap. Residues/sites (boxed) with cross-links (black solid lines) and electrostatic surface potentials indicated. Disease-related and de novo (d.n.) mutated residues indicated by dots coloured to match bars and legend colours in panel h. **h** In vitro assembly/solubility assay. SDS-PAGE of input (Inp.), soluble (S), and pellet (P) after centrifugation of wild-type and mutant lamins in oligomerisation buffer. Band intensities were quantified for relative soluble:pellet ratios; graph, n = 3 independent replicas, dots ratios in replicas, bars median replica values, error bars standard errors, p-values (T-test) for different from WT: R7/8 G 0.0076(**), E25G 0.0019(**), E33G 0.0139(*), E65G 0.00097(***), E82K 0.0588(n.s.), R388C 0.3054(n.s.), R388E 0.0072(**), R401C 0.1805(n. s.), R401E 0.0050(**). Mutations disrupting the interface increase the fraction of soluble material. **i** Frequency distribution of free lamin A dimers and dimers incorporated into head-to-tail oligomers determined by rotary metal shadowing EM reveals a similar effect of the mutations blocking the appearance of oligomers; n = 1000 per mutant. Black dashed lines in all structures schematically connect otherwise directly continuous protein fragments. Source data for panels **h** and **i** are available in the Source Data file

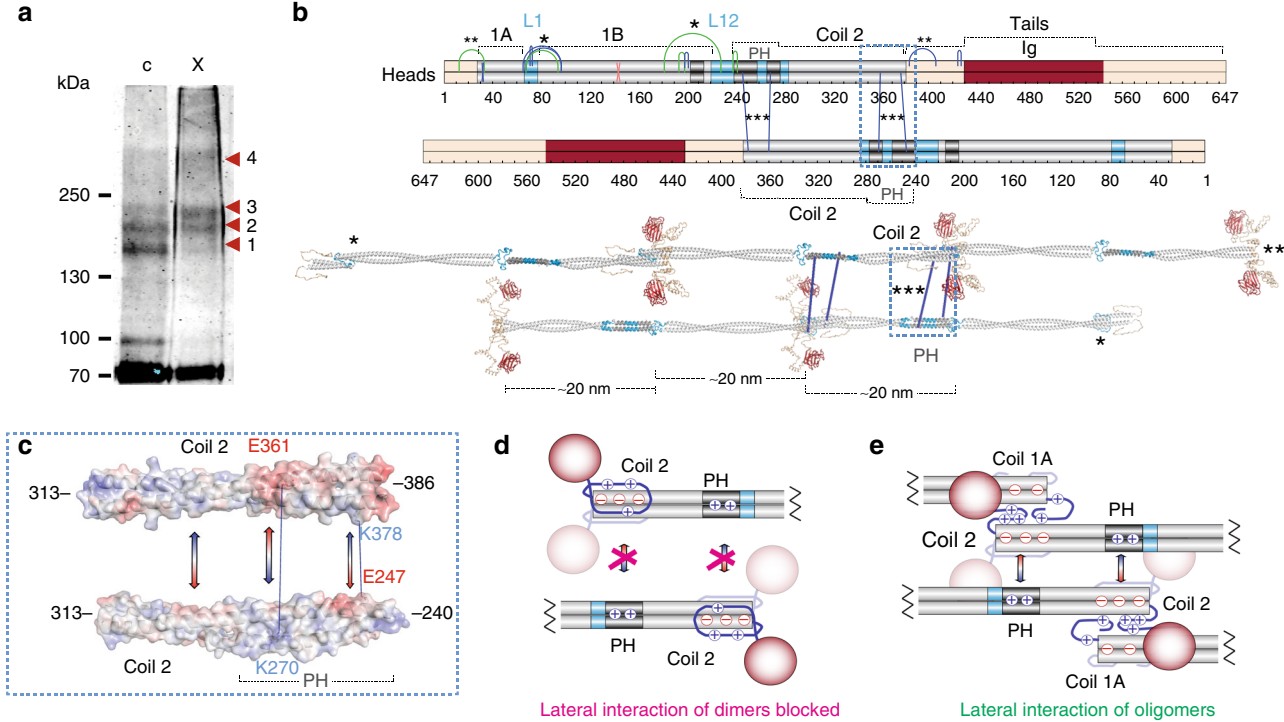

**Fig. 9** Ex vivo nucleoskeleton cross-linking indicates lateral strand assembly. **a** Lamin A Western blot of uncross-linked and cross-linked rat liver nuclear envelopes. Bands 1–4 appear before cross-linking and changes in band intensities parallel in vitro cross-linking, attesting to successful cross-linking. **b** Fifteen cross-links (top schematic) identified ex vivo: five were not previously seen in vitro (green) and 10 directly recapitulated in vitro results. Previously in-vitro-encountered cross-links E65-K97 and E68-97 (asterisk) and an additional cross-link E68-S94 all suggest coil 1A and L1 folding over coil 1B, thus recapitulating in vitro conclusions. Similarly, a cross-link D192-S239 indicates L12 and PH folding over coil 1B (asterisk) again recapitulating in vitro findings. Folding of the head and tail onto the adjacent rod termini was recapitulated in cross-link S12-E33 and in the previously seen E385-S407 cross-link (double asterisk). Inter-dimeric K270-E361 and E247-K378 (triple asterisk) cross-links were consistent with a lateral anti-parallel half-staggered dimer interface in filaments (bottom model). One cross-link S143-E145 (red) was strictly intra-molecular in vitro, but was inter-molecular (between overlapping peptides) ex vivo. **c** Alignment of coil 2 fragments with electrostatic potential overlay: negative (red), positive (blue). Gradient-filled arrows indicate potential weak electrostatic attractions. **d**, **e** Differences in spectra recovered between dimeric (**d**) and tetrameric (**e**) species suggest that these electrostatic interactions are likely blocked by the tail unstructure regions between dimers, but possible between head-to-tail oligomers. Source data for panel **a** are available in the Source Data file

**Lamin compression in the intact polymer.** We also tested whether these in vitro measured interactions also occur in the intact nucleoskeleton by investigating ex vivo cross-links in rat liver nuclear envelopes. As a general support, the ex vivo cross-linked material yielded a lamin A banding pattern similar to in vitro (Fig. 9a and Supplementary Table 1). CLMS analysis of

ex vivo cross-linked material yielded a total of 15 cross-links, 10 of which were also identified in vitro, and collected data further recapitulated the described key in vitro findings. The proposed mechanism for rod shortening via rod/linker tandem stagger folding in L1 and L12 is supported by linker-to-coil cross-links E68-S94, E68-K97 (L1-to-coil 1B) and D192-S239 (L12-to-coil

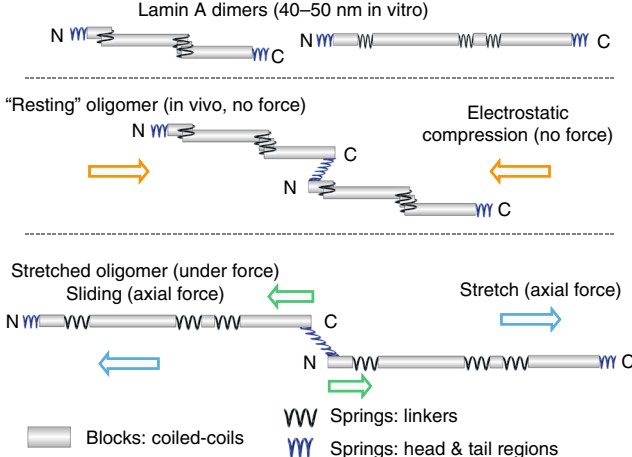

**Fig. 10** Model for lamin A compression spring. The lamin A dimer rod can compress to ~ 40 nm by the tandem staggering of coiled coils or be stable in a semi-relaxed state without the stagger at ~ 50 nm (uppermost schematics). In the assembled state disordered regions of the head and tail domains contribute a flexible connection with the rod at the tetrameric interface that can further compress the polymer in its resting state (middle schematic). Under tension stress these interactions are broken and the flexible regions can stretch so that they effectively act as springs that enable reversible dimer compression, stretching and sliding in the assembled lamin A polymer (bottom schematic)

1B) as well as E65-K97 cross-link between adjacent coils 1A and 1B termini in the assembled nucleoskeleton (Fig. 9b asterisk). Similarly, interactions of the unstructured head and tail regions flanking the rod were recapitulated in S12-E33 and E385-S407 cross-links (Fig. 9b double asterisk). Ex vivo cross-links also explain differences between lamins and other intermediate filaments in assembly: lamin dimers uniquely only engage in lateral assembly once incorporated into head-to-tail oligomers[14]. Cross-linking data suggest an antiparallel association of two dimers via their coils 2 (Fig. 9b triple asterisk), with the positively-charged PH region of one binding the acidic C-terminus of the other (Fig. 9c). In solitary dimers, however, the coil 2 acidic C-terminus is dynamically occupied by the flanking positively-charged unstructured tail region (Figs. 8c, 9d), and thus lateral association with the PH region would only be possible between head-to-tail oligomers where tail region interactions shift towards coil 1A (Figs. 8d, e and 9e). This explains earlier experiments where tail-less lamin dimer lateral interactions were inhibited by addition of this tail region[21].

## Discussion

These data indicate that electrostatic interactions between the flanking unstructured head and tail regions with the opposite rod termini of adjacent dimers establish and maintain the head-to-tail interface (Fig. 10). This function of the tail is unique to lamins as they are the only intermediate filaments with positive charge immediately after the rod: this added electrostatic interaction may help direct the order of assembly. By contrast, intermediate filament rod domain structure is more conserved, and its compression through linkers folding flanking coils over one another in a tandem stagger is likely a general property of all intermediate filaments. Strong pulling forces frequently exercised on lamins could break these interactions and extend the rod to increase polymer size via proximal re-organisation of electrostatic and polar interactions and stretching linkers. When force is removed, the unstructured linker and head/tail regions have multiple redundant pathways to re-establish the same compression and

assembly endpoints due to the range of possible electrostatic and polar interactions. Thus, linkers and unstructured head/tail regions likely act as springs enabling the stretch and compaction properties of intermediate filaments (Fig. 10). This mechanism can explain many previously inexplicable disease mutations in head/tail/linkers and adjacent coils and, as a strategy to generate elasticity, could be applied in synthetic polymer design.

## Methods

**Wild type and mutant lamin A expression vector cloning.** Lamin A without tags, its mutants and 6xHis-tagged lamin A were expressed in *E.coli* using pET28b vector (Novagen, #69865–3). Sequence encoding human pre-lamin A was PCR-amplified to exclude last 18 amino acids (to match the mature lamin A form as processed in a mammalian cell). An internal to lamin A Nco I site at position 1,388 was removed using site-directed mutagenesis and Nco I and BamH I sites were introduced to 3′ and 5′ termini for cloning of untagged wild type and mutant lamin A into pET28b deleating of the N-terminal His-tag. His-tagged lamin A was cloned using 3′ BamH I and 5′ Xho I instead. The following de novo (d.n.) lamin A mutants or mutants associated with Emery-Dreifuss Muscular Dystrophy (EDMD), Limb-Girdle Muscular Dystrophy (LGMD1B), Dilated Cardiomyopathy with Conduction Defects (DCM-CD) and Congenital Muscular Dystrophy (CMD) were used in this study: R7/8E double mutant (*d.n.*), R25G (EDMD/LGMD1B), E33G (EDMD/LGMD1B), E65G (EDMD/LGMD1B), E82K (DCM-CD), R388C (CMD), R388E (*d.n.*), R401C (EDMD) and R401E (*d.n.*). Corresponding mutations were introduced into the lamin A coding sequence using site-directed mutagenesis circular PCR of the pET28b-laminA vector. Full list of primers is available in Supplementary Table 2.

**SILAC lamin A expression and purification from bacteria.** Light and isotopically labelled heavy wild type lamin A were expressed in BL21 DE3 argA lysA cells[47] grown in M9 medium supplemented with 0.2% glucose, 50 µg mL$^{-1}$ kanamycin, 1 mM MgSO$_4$, and either normal or isotopically labelled L$-^{13}$C$_6$$^{15}$N$_4$-arginine and L$-^{13}$C$_6$$^{15}$N$_2$-lysine (50 µg mL$^{-1}$ each) (Sigma-Aldrich, #608033 and #608041). Overnight cultures were grown and diluted to an OD$_{600}$ of 0.001 and further grown for additional 8 h to an OD$_{600}$ of 0.6 before IPTG induction.

His-tagged lamin A was expressed in conventional BL21 DE3 bacterial cells in LB medium. In either case protein expression was induced with 0.3 mM IPTG for 4 h and protein was purified from inclusion bodies. Briefly, 1 L of bacteria were collected by centrifugation at 2,000 × $g$, lysed in 12 ml of 50 mM Hepes, pH 8.0, 3 mM MgCl$_2$, 3 mM 2-mercaptoethanol, 1 mM PMSF, 1 µg mL$^{-1}$ aprotinin, 1 µM leupeptin, and 1 µM pepstatin by repeated sonication in the presence of DNAseI (1 U µl$^{-1}$). Inclusion bodies were then pelleted, washed twice in ddH$_2$O with 2% Triton X-100 and resuspended in the buffer with 6 M urea[48]. Protein was equilibrated in either Tris buffer (25 mM Tris, 250 mM NaCl, pH 8.0) or NaPi buffer (100 mM sodium phosphate, 250 mM NaCl, pH = 8.0) supplemented with 6 M urea and 1 mM PMSF.

**Analytical ultracentrifugation.** For analytical ultracentrifugation lamin A was reconstituted immediately after purification in either Tris buffer or NaPi buffer with 6 M urea. Protein at a concentration of 0.5 mg mL$^{-1}$ was then dialysed against Tris or NaPi buffer without urea and prepared in 400 µl at a final concentration of 0.3 mg mL$^{-1}$. Sedimentation velocity (SV) experiments were carried out in a Beckman Coulter (Palo Alto, CA, USA) ProteomeLab XL-I analytical ultra-centrifuge using interference optics. All AUC runs were carried out at a rotation speed of 45,000 rpm (154,000 × $g$ at the optical cell centre) and an experimental temperature of 20 °C. The density and viscosity of the Tris buffer and NaPi buffer at the experimental temperature (20 °C) were calculated using program SEDN-TERP[49]. The partial specific volume ($\bar{v}_{20}$) of protein was calculated as an additive sum of values of constituent amino acids using program SEDNTERP. Sedimentation velocity profiles (separate scans were taken every 4 min, 140 scans in total) were treated using size-distribution c(s) model implemented in the program SEDFIT[50]. Each peak on the distribution plot was integrated in order to obtain the weight-averaged values for sedimentation coefficient and molecular mass. Integrated values of sedimentation coefficient (s) obtained at experimental conditions were converted to the standard conditions (s$_{20,w}$) (which is the value of sedimentation coefficient in water at 20 °C).

Sedimentation coefficients $S_{calc}$ for the lamin A dimeric models with different levels of the rod compression based on cross-linking data and lamin A head-to-tail tetramer mock models were calculated from their atomic coordinates using program SoMo[51,52].

**Lamin homo-iso- and homo-/hetero-iso-dimer mix preparation.** Homo-iso-dimer mix of light and heavy lamin A was prepared by mixing light and iso-topically labelled heavy lamin A at a 1:1 ratio in NaPi buffer in which we have shown that lamin dimers do not exchange chains (Supplementary Fig. 2).

Homo-/hetero-iso-dimer mix was prepared by running the homo-iso-dimer mix on a preparative Hoefer 180 mm × 160 mm 10% Laemmli SDS-PAGE gel. A

total of 1.6 mg of protein per a single slab was run for 8 h at 150 V. The area containing monomerised and mixed-in-the-gel light and heavy lamin A was identified via IntstantBlue (Expedeon, #ISB1L) staining of a thin strip of the gel. Unstained protein was then gel-extracted and desalted[28,29] via precipitation with 400 mM KCl and sequential washes with an 86:7:7 (vol:vol) mix of Acetone, Triethylamine and Acetic Acid and the same solution diluted to 5% in ddH$_2$O[48]; then and reconstituted in NaPi buffer with 6 M urea. Successful refolding of gel-extracted lamin A was confirmed by rotary metal shadowing EM. As light and heavy lamin A dimers were monomerised together on an SDS-PAGE in this procedure, refolded protein sample should contain a combinatorial mix of 25% light-light, 25% heavy-heavy and 50% light-heavy lamin A dimers.

**Lamin A in vitro cross-linking and mass spectrometry**. Cross-linking of lamin A in vitro was carried out at a protein concentration of 0.4 mg mL$^{-1}$ in NaPi buffer: Homo-iso-dimer or homo-/hetero-iso-dimer lamin A mixes were dialysed into NaPi buffer at a concentration of 0.5 mg mL$^{-1}$ and adjusted to 0.4 mg mL$^{-1}$ in each sample. This concentration was chosen as one close to maximum concentration at which lamin A can stay soluble in the chosen buffer for the duration of the cross-linking experiment.

Samples prepared in this manner were then cross-linked with a mixture of EDC (1-ethyl-3-(3-dimethylaminopropyl)carbodiimide hydrochloride) and Sulfo-NHS (N-hydroxysulfosuccinimide) (Thermo Fisher Scientific Pierce, #PG82074 and # 24510) at a weight ratio of 1:4:8.8 (lamin A:EDC:S-NHS) for 30 min at room temperature. The reaction was then quenched for 10 min by addition of 1 M Tris at pH 8.0 to a final concentration of 50 mM. For SDS-PAGE analysis the quenched cross-linking reaction was split in two halves and resolved on a preparative poly acrylamide gels: 7.5% Bis-Tris gel for mass spectrometry analysis, or a 7.5% Laemmli gel for gel-extraction of individual bands.

For MS analysis, polyacrylamide gels with resolved cross-linking reaction product-bands were stained with InstantBlue and de-stained with ddH$_2$O before band excision.

The bands corresponding to cross-linked complexes were excised and the proteins therein were reduced using 10 mM DTT for 30 min at room temperature, alkylated with 55 mM iodoacetamide for 20 min in the dark at room temperature and digested using 13 ng μL$^{-1}$ trypsin (Thermo Fisher Scientific) overnight at 37 °C[53] and digested peptides were fractionated using SCX-Stage-Tips[54]. In short, peptide mixtures were first loaded on a SCX-Stage-Tip in 0.5% (v/v) acetic acid, 20% (v/v) acetonitrile, 50 mM ammonium acetate and sequentially eluted with buffers containing 100 mM ammonium acetate and 500 mM ammonium acetate (two fractions each). Each peptide fraction was then desalted using C18-Stage-Tips[55,56] prior to mass spectrometric analysis.

LC-MS/MS analysis of peptides in the in vitro cross-linking experiments was performed on an LTQ Orbitrap Velos mass spectrometer (Thermo Fisher Scientific) that was coupled with a Dionex Ultimate 3000RSLC nano HPLC system, using a high/high strategy[57], both MS and MS2 spectra were acquired in the Orbitrap. The analytical column with a self-assembled particle frit[54] and C18 material (ReproSil-Pur C18-AQ 3 μm; Dr. Maisch, GmbH) was packed into a spray emitter (75-μm ID, 8-μm opening, 300-mm length; New Objective) using an air-pressure pump (Proxeon Biosystems). Mobile phase A consisted of water with 0.1% formic acid. Mobile phase B consisted of 80% acetonitrile with 0.1% formic acid. Peptides were loaded onto the column with 2% B at 500 nl min$^{-1}$ flow rate and eluted at 200 nl min$^{-1}$ flow rate, with a linear gradient increased from 2 to 40% acetonitrile in 0.1% formic acid in 139 min and then an increase from 40 to 95% B in 11 min. Mass spectra were recorded at 100,000 resolution. The eight highest intensity peaks with a charge state of three or higher were selected in each cycle for ion-trap fragmentation. The fragments were produced using collision-induced dissociation (CID) with 35% normalised collision energy and detected by the Orbitrap at 7500 resolution. Dynamic exclusion was set to single repeat count and 90 s exclusion duration. The mass spectrometric raw data were processed to generate peak lists by MaxQuant (Version 1.5.3.30)[58] and cross-linked peptides were matched to spectra by Xi software (version 1.6.745)[59] with in-search assignment of monoisotopic peaks[60] and the following parameters: sequence database human LaminA; cross-linker: EDC; MS accuracy, 6 ppm; MS/MS accuracy, 20 ppm; enzyme: trypsin; missed cleavages, 4; missing mono-isotopic peaks: 4; fixed modification: carbamidomethylation on cysteine; variable modifications: oxidation on methionine; R10 and K8 for SILAC samples. Search database is human lamin A with decoy setting. FDR was estimated using XiFDR (version 1.1.27) on 5% residue level[61].

**Lamin A ex vivo cross-linking and mass spectrometry**. Rat liver nuclear envelopes were purified according to standard procedures developed in Schirmer lab[62]. Livers were obtained from Sprague Dawley rats provided by the University of Edinburgh animal facility in compliance with local ethics and Home Office procedures. Liver tissue was homogenised and nuclei were isolated and stripped of the remaining endoplasmic reticulum and contaminants in a series of centrifugation spins through sucrose cushions of increasing concentrations. Intact chromatin still inside the nuclei was then digested with DNAse and micrococcal nuclease (MNAse) and washed away from the nuclei in a hypotonic buffer as confirmed by means of Hoechst staining that revealed only a thin rim of chromatin still clinging to the nucleoskeleton. Purified nuclear envelopes were then pelleted and

equilibrated in NaPi buffer and cross-linked with a 1:2.2 mixture of EDC/S-NHS supplied in twofold and fourfold NE protein weight excess for 30 min. Reactions were quenched with 50 mM Tris pH 8.0 for 10 min. Optionally purified nuclear envelopes were additionally extracted with 0.5% Triton X-100 and 400 mM KCl to obtain lamina-pore complex nuclear envelope shells[63].

Successful nucleoskeletal lamin A cross-linking was confirmed by Western blotting of cross-linked nuclear envelopes lysed in NuPage LDS buffer (Thermo Fisher Scientific) and resolved on an 8% Bis-Tris polyacrylamide gel with the 5881 rabbit polyclonal antibody[64] against a region [572–585] downstream of the Ig fold in the lamin A tail domain (at a 1:250 dilution). LICOR anti-rabbit IR800 antibody (Li-Cor Biosciences, #925–32213, 1:2,500 dilution) was used as secondary. Imaging was done with a Li-Cor Odyssey CLx imaging system (Li-Cor Biosciences).

In preparation for MS analysis cross-linked nuclear envelopes were washed in 1 M NaCl and the lamina was solubilised in 0.1 M Tris 8.5, 4 M urea, 20 mM MgCl$_2$. Around 10% of total nuclear envelope protein was recovered this way and then processed for mass spectrometry by means of sequential overnight digestion with endoproteinase Lys-C (Roche, # 11058533103) dilution to 2 M urea and overnight digestion with Trypsin (Thermo Fisher Scientific, # 90057). Alternatively in-gel[53] digestion and FASP digestion[65] was used. In either case digested peptides were fractionated using SCX-Stage-Tips as described in the in vitro cross-linking method section.

LC-MS/MS analysis of peptides in ex vivo cross-linking experiments was performed using either an LTQ Orbitrap Velos mass spectrometer (Thermo Fisher Scientific, details see above) or an Orbitrap Fusion™ Lumos™ Tribrid™ Mass Spectrometer (Thermo Scientific) (as specified in the raw MS data files) applying a "high-high" acquisition strategy. Peptides were separated on a 75 μm × 50 cm PepMap EASY-Spray column (Thermo Scientific) fitted into an EASY-Spray source (Thermo Scientific), operated at 50 °C column temperature. The eluted peptides were directly introduced into the mass spectrometer. MS data were acquired in the data-dependent mode with the top-speed option. For each three-second acquisition cycle, the survey level spectrum was recorded in the Orbitrap with a resolution of 120,000. The ions with a precursor charge state between 3+ and 8+ were isolated and fragmented using high energy collision dissociation (HCD) of 30% normalised collision energy. The fragmentation spectra were recorded in the Orbitrap with a resolution of 15,000, isolation window of 1.6 m/z, maxium injection time of 60 ms and AGC target of 5e4. Dynamic exclusion was enabled with single repeat count and 60 s exclusion duration. The mass spectrometric raw data were processed to generate peak lists by MSCovert (ProteoWizard 3.0.11417)[66] and cross-linked peptides were matched to spectra using Xi software (version 1.6.745)[59]. The parameters are the same as for in vitro data analysis except for using a database of rat lamin A instead of human. The data has been manually validated.

All the mass spectrometry proteomics data have been deposited to the ProteomeXchange Consortium via the PRIDE[67] partner repository with the dataset identifier PXD008337 and PXD014009.

**SILAC cross-linking data analysis**. A total of 1,382 cross-link spectra were identified in dimer-rich bands 1–3 in the homo-iso-dimer mix (HIDm X) experiment after FDR using XiSearch engine (version 1.6.745)[59] and XiFDR (v 1.1.27)[61]. Similarly 1.322 cross-link spectra were identified in dimer-rich bands 1–3 in the homo-/hetero-iso-dimer experiment (H/hIDm X). Identified spectra of peptide pairs with incomplete SILAC labelling were discarded. To increase the stringency of our analysis only spectra with match_score below 8.0 were also discarded, leaving 807 and 858 spectra (1,665 total) supporting 387 and 365 unique cross-linked residue pairs in the respective cross-linking experiments. These two sets overlapped in 243 unique cross-linked residue pairs—cross-links identified in both experiments. These were supported by 1308 spectra out of total 1665 (78%) attesting to a high reproducibility of abundant cross-links.

To determine the inter-/intra-dimeric and inter-/intra-chain origin of each of the 243 cross-links a comparative cross-linking analysis[30,31] routine was adapted (Supplementary Discussion): for each of the 1.308 spectra relative ion intensities in four peptide-pair precursor ion clusters—light-light peptide-pair ion cluster (LL), heavy-light (HL), light-heavy (LH) and heavy-heavy (Fig. 2e–g)—were identified and quantified and quantified across the pertinent elution peak (extracted ion chromatograms XiCs) containing this spectra MS and MS:MS. Due to the extreme complexity of the sample these were quantified manually in Thermo Xcalibur software (Thermo Fisher Scientific 4.0.27.1) as average peak intensities across pertinent XiCs. For a select set of spectra XiC areas were also quantified semi-automatically using Skyline[32,68] (v 3.7.0.11317). In each case mono-, first and second isotopic peak XiCs were analysed in each of 4 peptide ion clusters. Single XiC was used for all peaks in Xcalibur and individual XiCs were aligned manually in Skyline. Relative HL and LH peptide pair ion abundances were quantified as ratios (HL + LH)/(LL + HL + LH + HH) for each spectra. Then frequencies of cross-link occurrence between differentially labelled peptides were calculated using as median values of (HL + LH)/(LL + HL + LH + HH) ratios of all spectra supporting this cross-link in HIDm X and H/hIDm X experiments separately (Supplementary Data 4). Ratios calculated using Xcalibur and Skyline appeared to match closely and lay within 6% of each other for 95% of measurements (Supplementary Data 1–3). These were converted into frequencies F$^{Inter-dimeric}$ F$^{Inter-molecular}$—of cross-link occurrence respectively as inter-dimeric in the HIDm X experiment and as inter-chain in the H/hIDm X experiment (Fig. 1g). Respective absolute frequencies of cross-link

occurrence as intra-dimeric-inter-chain ($F^{Inter-chain}$) and intra-chain ($F^{Intra-chain}$) were then calculated as $F^{Inter-chain} = F^{Inter-molecular} - F^{Inter-dimeric}$ and $F^{Intra-chain} = 1 - F^{Inter-chain} - F^{Inter-dimeric} = 1 - F^{Inter-molecular}$. A number of cross-links have shown a propensity to happen both between and within dimers and both between and within chains owning to two factors: on the one hand, the extreme flexibility and assembly/aggregation tendency of lamins; and on the other, the fact that majority residues in the rod domain can be considered as surface residues and thus can engage in both intra- and inter-molecular interactions. For each cross-linked region of lamin A we aimed to reconstruct its frequent conformation(s). To this end during consequent interrogation of the obtained cross-links for structural information the following arbitrary interpretation of the $F^{Inter-dimeric}$, $F^{Inter-chain}$ and $F^{Intra-chain}$ were in effect: a cross-link with the $F^{Inter-dimeric} \geq 80\%$ was considered to be mostly inter-dimeric and not considered for lamin A dimer structure; within the remaining pool of cross-link each was then considered intra-chain in a dimer if the $F^{Inter-chain}/(F^{Intra-chain} + F^{Inter-chain}) < 20\%$ (among all dimers containing this cross-link, in 80% or more cases this cross-link happens within a single chain); considered inter-chain in a dimer if $F^{Inter-chain}/(F^{Intra-chain} + F^{Inter-chain}) \geq 80\%$ (happens between the two chains in 80% or more cases); considered to happen both ways—intra-chain and inter-chain in a dimer if $F^{Inter-chain}/(F^{Intra-chain} + F^{Inter-chain})$ was between 20 and 80%.

Additionally, 220 spectra supporting 143 unique cross-linked residue pairs were identified in the tetramer-rich band 4 in the homo-iso-dimer mix experiment. These were similarly processed to obtain ($F^{Inter-dimeric}$).

Based on this information it was possible to assess fitness of the cross-links to the prediction based model of lamin A.

**Ex vivo cross-linking data analysis**. Thirty nine spectra supporting 15 unique cross-linked residue pairs were identified from multiple ex vivo cross-linking experiments. Two cross-links—E31-K32 and S143-E145—were unambiguously identified as inter-molecular as these occurred between pairs of overlapping peptides. Remaining cross-links were presumed as inter- or intra-molecular based on in vitro data.

**Cross-linking data visualisation**. Cross-linking data from all experiments were visualised using xiNET (http://crosslinkviewer.org/index.php)[69].

**Cross-linked protein isolation via gel-extraction**. To be able to relate cross-linking patterns obtained from each cross-linking product-band to the lamin A oligomeric architecture in the same product-band the second half of the reaction was resolved on 7.5% Laemmli SDS-PAGE. Upon resolution a thin band was cut to identify the exact migration pattern and individual bands were excised, gel-extracted, desalted and reconstituted as described before for the homo-/hetero-iso-dimer lamin A mix.

**Rotary metal shadowing EM**. For rotary metal shadowing EM of un-cross-linked lamin A, the protein was equilibrated in either Tris buffer or NaPi buffer at a concentration of 0.1 mg mL$^{-1}$ by dialysis against 1,000 volumes twice. For rotary metal shadowing of cross-linked lamin A and material in the individual cross-linked bands, protein in each sample was equilibrated in NaPi buffer at the same concentration again by dialysis. Glycerol was then added to the equilibrated lamin A samples to a final concentration of 30% and samples were sprayed onto roughly square $10 \times 10$ mm mica sheet pieces (TAAB, #M460) using a glycerol spraying device[26] graciously provided by Prof. Ueli Aebi. These were then rotary metal shadowed in an ACE600 Leica vacuum evaporator at a pressure of $1.0-2.5 \times 10^{-5}$ mbar with 2 nm of platinum and 9 nm of carbon at respectively 6° and 90° elevation angles. Coated mica sheet pieces were then incubated in a moist chamber at 60 °C for 30 min and platinum/carbon cast films were floated on water. Pieces of these films were picked with EM copper grids and air-dried. Imaging was performed using JEOL 1200 TEM at 80 kV and magnification of ×20,000. Micrographs were then analysed in ImageJ and lamin A rod length was recorded. Distributions were not assumed normal, and non-parametric Kruskal-Wallis test followed by pair-wise comparison Dunn test with Holm correction were carried out to estimate shift in rod length distributions.

**Basic lamin A dimer modelling**. Basic model of lamin A dimer with 51 nm long rod was built using existing structures, homology modelling and de novo modelling. Coil 1A, part of L1 and the bulk of coil 1B were modelled de novo using CCbuilder (v 1.0)[70]. Residues [181–220] in the C-terminus of coil 1B were modelled by homology with vimentin: 3UF1 [https://doi.org/10.2210/pdb3UF1/pdb] structure[34] fragment containing residues [208–247] was used as a template for SWISS-MODEL (https://swissmodel.expasy.org/)[71–74]. Similarly, the N-terminal half of coil 2 containing a parallel hendecoad (PH) residues [240–277] and coil 2 residues [288–310] were modelled by homology with vimentin using 3TRT [https://doi.org/10.2210/pdb3TRT/pdb] structure[35] fragment containing residues [264–334] as a template for SWISS-MODEL. The C-terminal half of coil 2—residues [313–386]—was directly copied from $1 \times 8Y$[https://doi.org/10.2210/pdb1X8Y/pdb] crystal structure[8]. Missing residues in the rod were built using modelled fragments as a single template for SWISS-MODEL. N-terminal head and

the tail domains were modelled using iTasser[75–77]. Residues of the Ig fold were then substituted by superposition of the 1IFR [https://doi.org/10.2210/pdb1IFR/pdb] crystal structure[10]. All further manipulation of the disordered regions to build projection models in Fig. 8 and Supplementary Figs. 7 and 9 were done in PyMol[78].

**Molecular modelling of the lamin A head-to-tail tetramers**. To calculate theoretical sedimentations coefficient $S_{calc}$ for lamin A head-to-tail tetramers, mock tetrameric models were created using Rosie docking2 protocol—part of the Rosetta online public server[79–81]. Coil 2 fragment [354–385] was roughly aligned to coil 1A fragment [28–67] in a half-staggered manner as an input for docking2 to produce models with 2-3-heptad-long interfaces thought to occur in during lamin head-to-tail tetramerisation[24]. Alternatively longer fragments [320–385] and [28–90] were used to produce models with interface spanning the entire length of coil 1A. Among top 10 scoring models those with roughly parallel coiled coil fragment alignment were selected and rest of the dimers' rod and unstructured region were build using the basic lamin A dimer model (see section on Basic lamin A dimer modelling) in PyMol.

**Molecular modelling of the rod coiled coil segment overlap**. To model lamin A coiled coil termini overlap and rod compression from available coil-to-coil cross-links across linkers L1, L12, L2 and L3 cross-link guided molecular modelling was employed[43,82] that utilises ROSETTA (build 2017.08.59291) global docking protocols[83] and Xwalk (v 0.6)[44] to select structures satisfying input cross-links. Rosetta and Xwalk command parameters are detailed in Supplementary Note 1.

Here linkers were assumed flexible and thus only cross-links between coiled coil termini were used as an input (Supplementary Data 5 and 6). Intra- and inter-molecular cross-links were used in duplicates (i.e. respectively from chain A to chain A and from chain B to chain B; from chain A to chain B and from chain B to chain A), cross-links that happen both ways were used in tetraplets (A–A, A–B, B–A, B–B) Docking was performed between dimeric coiled coil fragments [46–66] and [78–102] separated by L1, [201–219] and [240–260] separated by L12, [241–254] and [267–291] assumed separated by (L2), and [256–276] and [289–310] assumed separated (L3). A total of 100,000 models were generated for each pair of fragments.

Models with negative I_sc (Rosetta's Interface score calculated as a difference between the total energy of the complex and the sum of total energies of each partner in isolation; models with I_sc < −5.0 are considered as good decoys as detailed in RosettaDock application documentation https://www.rosettacommons.org/docs) were chosen and surface accessible distances ($C_{\beta}$-$C_{\beta}$ SAS distances) between pairs of $C_{\beta}$ atoms of cross-linked residues were calculated with Xwalk (-xSC option) and in some cases manually (Supplementary Note 1).

Models failing to satisfy any cross-links were discarded. Angles between docked dimeric fragments were calculated in PyMol using angle_between_helices function from psico module (by Thomas Holder and Steffen Schmidt, https://github.com/speleo3/pymol-psico). Of the remaining models only those with two dimeric fragments in parallel or close to parallel orientation (tandem stagger models) were picked for further analysis, while the rest (anti-parallel folds) were discarded. Tandem stagger models were additionally refined using Rosetta local refinement protocols to achieve better interface scores. Cross-links were re-validated with Xwalk and tandem stagger structures with negative I_sc, satisfying cross-links were included into a final set (Supplementary Data 7–11). Rod shortening conveyed by each model was then calculated in PyMol as change in distance between the straight dimer rod length and the rod length in each model, calculated as a sum of rod fragments before and after the tandem stagger point and thus taking in consideration the bending of the rod introduced in the stagger. Finally models with I_sc ≤ −5.0 were deemed as those with stable interface between docked dimeric fragments.

In further search for a more exhaustive list of tandem stagger models Rosetta was run in an unconstrained mode—with no cross-links as an input—for each pair of dimeric coiled coil fragments separated by linker L1, L12, (L2) and (L3). Models with negative I_sc were initially filtered with Xwalk for those roughly satisfying cross-links (as defined by maximal $C_{\beta}$-$C_{\beta}$ SAS distances for cross-linked residues of 13Å). Any models potentially satisfying at least one cross-link were then refined and models with I_sc above −5.0 were discarded. Remaining models were filtered for compliance with maximal extension allowed by the respective linkers L1, L2 and L3 (SAS 46 Å S66O-K78N, SAS 46 Å D254O-Y267N and SAS 46 Å Q276O-H289N) via Xwalk. The linker L12 is sufficiently long to accommodate any degree of overlap dictated by coil 1B-to-PH cross-links and is altogether longer than maximal SAS distance of 60 Å that Xwalk can calculate; therefore filtering for compliance with L12 extension was redundant. Remaining models were then subject to a more stringent test for cross-link satisfaction via Xwalk in search for models where cross-linked residues already engage in electrostatic or polar interaction with each other. To this end instead of $C_{\beta}$-$C_{\beta}$ SAS distances straight line Euclidean distances were calculated between carboxyl-group oxygen atoms and side chain amine group nitrogen atoms (for lysine-involving cross-links) or hydroxyl group oxygen atoms (for serine, tyrosine or threonine involving cross-links). Residues were then assumed cross-linkable if these distances did not exceed 4 Å and thus satisfying close range electrostatic interaction criteria of 4 Å between oxygen and nitrogen atoms in COO- and $CH_2NH_3$ + (for D/E-K)[84] or hydrogen

bond criteria (for D/E-Y/T/S)[85]. Angles between docked fragments were calculated in PyMol and tandem stagger models were selected.

**Linker regions reconstruction for hydrodynamic calculations**. To carry out hydrodynamic calculations and determine sedimentation coefficients of lamin A rod in various stages of compression, docked dimeric fragments from Z-fold models were incorporated into the basic lamin A dimer model and missing linker regions were reconstructed using SWISS-MODEL or MODELLER[86]. Input commands for the latter are detailed in Supplementary Note 1.

**Electrostatic potential surface (EPS) reconstruction**. EPS reconstructions were carried out for coil 1A, coil 2 C-terminus and head and part of the tail before the Ig domain using PDB2PQR (v 2.1.1) servers (http://nbcr-222.ucsd.edu/pdb2pqr_2.1.1/)[87] in Amber force field at pH 8.0. Output.pqr files were then computed in APBS using linearized Poisson–Boltzmann Equation; mobile ions were set to match salt concentration in the cross-linking buffer. All relevant Figures feature EPS with 50% transparency as visualised in PyMol.

**In vitro assembly assays**. Reconstituted lamin A and its mutants were first equilibrated in 25 mM Tris, 8 M urea, 250 mM NaCl, pH 8.8 at a concentration of 0.3 mg mL$^{-1}$ and then dialysed out of urea against 25 mM Tris, 250 mM NaCl, 1 mM DTT, 1 mM EGTA, pH 8.5 for 3 h and then against 25 mM MES, 150 mM NaCl, 1 mM DTT, 1 mM EGTA pH 6.5 for an additional 30 min. For the general solubility assessment, samples were then centrifuged at $100,000 \times g$ for 30 min, supernatant was immediately removed as soluble phase (S) and an equal volume of 25 mM Tris, 8 M urea, 250 mM NaCl, pH 8.8 was added to the centrifuge tubes to solubilise pelleted lamin A oligomers (P). The experiment was carried out in biological triplicate, normal distribution was assumed and statistical significance was estimated in T-test comparison of mutant samples to the control sample. For a more detailed assessment of lamin A head-to-tail oligomerisation after dialysis, samples were supplemented with glycerol to a final concentration of 30% and processed for rotary metal shadowing as described above. Lamin A dimers were then scored as singular or incorporated into head-to-tail oligomers in ImageJ.

**Reporting summary**. Further information on research design is available in the Nature Research Reporting Summary linked to this article.

## Data availability

All the mass spectrometry proteomics data generated in this study have been deposited to the ProteomeXchange Consortium via the PRIDE[67] partner repository with the dataset identifiers PXD008337 and PXD014009. Rosetta molecular modelling data is available via Edinburgh DataShare (https://datashare.is.ed.ac.uk/handle/10283/3348). The source data underlying Figs. 1c–f, 2a, c, 7c, 8h, i, 9a and Supplementary Figs. 1a, b, 2a, and 6 are provided as a Source Data file. All other data supporting the findings of this study are available from the corresponding authors upon reasonable request.

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

## Acknowledgements

We thank Ueli Aebi for assistance setting up rotary shadowing, Zhuo Angel Chen for SILAC help, and Bill Earnshaw for critical input. This work was supported by Wellcome Senior Research Fellowships 095209 (E.C.S.) and 103139 (J.R.) and Wellcome Centre for Cell Biology core grant 092076. A.A.M. was supported by a University of Edinburgh Principal's PhD Scholarship.

## Author contributions

Conceived/ designed experiments: A.A.M., E.C.S., J.R. Performed experiments: A.A.M., J.Z., D.R.H., C.S., A.S.S., C.C.P. Provided reagents/analysis tools: J.R., A.S.S. Wrote the manuscript: A.A.M., E.C.S.

## Additional information

**Competing interests:** The authors declare no competing interests.

