## [Peer Review File · Nature Communications]

Reviewers' comments:

Reviewer #1 (Remarks to the Author):

In their manuscript „lamin A molecular compression and sliding as mechanisms behind nucleoskeleton elasticity“ Makarov et al. investigate how lamin A self-assembles into a filamentous nucleoskeleton polymer. Using mainly rotary shadowing EM and a very thorough and comprehensive analysis by crosslinking mass spectrometry together with SILAC labelling the authors can demonstrate how the intrinsically disordered head, tail and linker regions are likely connected to each other in different lamin A complexes and how this relates to the overall arrangement, assembly and mechanics of the nucleoskeletal filament.

Apart from the fact that the manuscript is very densely written and thus at times very difficult to follow, it contains a large number of thoroughly executed experiments leading to novel and -as far as the reviewer can judge - biologically relevant information, that allow the authors to propose a novel mechanism of molecular compression and sliding being the cause of lamin A higher order assembly.

Given the biological relevance of this nucleoskeletal filament in health and disease, the work by Makarov et al., certainly deserves publication in Nature Communications.

However, some points, in particularly addressing the validation of the workflow used in this study to detect SILAC labelled crosslinks, need to be addressed before the manuscript is ready for publication.

Comments:

1. The adaption of the particular search engine used in this manuscript to detect SILAC labelled proteins should be validated using a well-established homo-dimer that is known to more easily associate and dissociate than Lamin A, preferably using all variations as input (e.g. both proteins non-labelled, both proteins SILAC labelled and labelled/ not - labelled).

At the very least it should be tested how many false positive hits the workflow is generating if no SILAC labelling is used - so using non-labelled lamin A but searching the data using the SILAC settings (e.g. allowing R10 and K8 as variable modifications).

2 The authors should demonstrate that after gel extraction and refolding hetero-iso-dimers (dimers consisting of one heavy chain and one light chain) are indeed formed.

In the extended data figure 1a the authors show that after incubation of untagged and His-tagged lamin A the untagged lamin A elutes already in the wash fraction from the nickel column and not together with His-tagged lamin A in the eluate fraction containing 250 mM imidazole. Thus, in order to really prove that hetero-iso-dimers are indeed formed under the conditions used (e.g. gel extraction and refolding) – this experiment should be repeated with gel extracted and refolded untagged lamin A and His-tagged lamin A to show that after gel extraction and refolding untagged lamin A co-elutes with His-tagged lamin A from the nickel column.

Minor Points:

1 For the definition of the Inter-chain / inter-molecular / inter-dimeric ratios in the supplementary discussion the authors mention: “likewise, in the second experiment (...) a crosslink occurring only between similarly labelled pairs of peptides would be a cross-link happening strictly within a single molecule of lamin A”.

Could in such an experimental setting a crosslink with similarly labelled peptides (e.g. L/L or H/H) not also originate from different lamin A molecules (either different molecules of a dimer or different dimers)? Maybe the authors could comment on this.

2 For reasons of completeness and if easily done, it might be nice if the elution peak areas of the 4 peptide-pair precursor ion clusters used for the manual quantification in Xcalibur? (The 'LFQ' values to calculate the ratios) could also be provided (and not only the ratios).

3 What is meant by „Z-folding“ of the linker?

4 In line 91 in the text it is stated that 'EDC cross-linking captured rod shortening (Fig. 1 c: X, 1-3) ...' it seems the text should refer to Fig 1f?

5 Would it be possible to strengthen the point made from the gel in figure 1f by directly measuring the length of the dimers by rotary shadowing EM to directly demonstrate that the different crosslink bands 1, 2 and 3 contain dimer fractions of a preferred length? E.g. dimers with a length of 40-44 nm are significantly enriched in band 1, whereas band 3 contains more dimers with a length of 48-50 nm.

6 What is the definition of “short range” and “long range” crosslinks?

7 With the exception of Rosetta, the exact software versions of the used software programs are missing and should be added and the settings used for modelling should be explained.

8 It would also be highly advisable to re-run at least some of the Rosetta runs using only a subset of the crosslinks as input to assess the robustness of the obtained models.

9 It might be a good idea to deposit the basic lamin A model, the Rosetta output (as well as the refined output) and the code snippets inside a publicly available git repository. This is especially true for the Modeller code since Python code is sensitive to white spaces and might be changed by the print formatting.

10 According to the supplementary information crosslinks were categorized and validated manually as well as via Xwalk. It would be nice to add a table showing the results of this validation as well as the calculated crosslink distances.

Reviewer #2 (Remarks to the Author):

The manuscript reports on the use of chemical crosslinking towards elucidating the structure of nuclear lamins. In vitro assembled lamins have been studied first, revealing the structure of the dimer as the dominant solution species. In addition (and somewhat surprising since the conditions used are those supporting soluble dimers rather than further assembly), data on the head-to-tail contact of the dimers have been obtained. This contact reveals critical interactions of the positively charged head and tail regions with the negatively charged rod ends, in line with existing predictions. In addition, some limited crosslinking data have been collected on ex-vivo lamina.

Chemical crosslinking has to my knowledge not been reported for lamins before, and the wealth of obtained crosslinking data (generally sound, with some reservations mentioned below) is the main strength of the manuscript. The conclusions about the possible ‘Z-folding’ of selected parts of the dimeric rod are novel and generally appearing worthy, while the biological significance still largely remains unclear, beyond some speculations provided.

The manuscript has apparently been initially written for a different journal and contains, first, an extremely condensed main text and main figures, followed by a set of supplementary figures (with lengthy legends, discouraged in Nature Comms), methods and supplementary discussion on crosslinking approach. I found this layout extremely cumbersome. The main text includes many conclusions that sound totally unconvincing unless the reader works own and hard way through all additional sections. Getting a clue on the crosslinking approach used is impossible without reading the supplementary figure legends and also the (very useful) supplementary discussion. A figure encompassing all crosslinks within the dimer is nowhere to find; to get an idea of just the intra-rod crosslinks, one has to examine Fig. 2b, then extended data Fig. 2a and also extended data Fig. 2f!!

The very beginning of the main text is quite unfortunate. The authors present some data such as rotary shadowing and UC towards a variability of the dimer length and its shortening upon crosslinking. These data are neither too convincing nor telling. Next, while the 20nm periodicity of the lamin filaments was seen in recent cryo-EM work on native lamina, discussing the 'in vitro' 25nm periodicity (seen in paracrystals, possibly?) is not too relevant.

Most likely the interesting crosslinking data and the Z-folding models have been the starting point of all these questions, and they should be reported first. The in vitro variability of the dimer (a topic with limited applicability to the native situation anyway) could be discussed thereafter as providing some (limited) support to the Z-folding possibility.

Overall the current manuscript is too lengthy and should be best split in two separate manuscripts IMO.

1. The first manuscript could focus on in vitro crosslinking (the strongest part of this work as mentioned) and dimer/ H2T tetramer modelling. Fully detailed account of the original crosslinking and modelling approaches, presented logically without the need to hunt through endless supplements, would be an interesting reading.

I have a bit of concern about the validity of deciding on the type of crosslink (intrachain, intradimer or interdimer) from the heights of MS peaks. It is known that the latter are often not representative of the relative abundance of different peptides, which may 'fly differently' in MS.

In addition, I wonder if an independent proof of the resulting structural models can be obtained using a very different chemical crosslinking strategy, such as with the classical BS3 for instance?

Nearly all crosslinks used to produce the Z-folded models (Fig. 2b) appear to be both intrachain and intradimeric. Is there a logical reason for this?

2. Second manuscript could focus on crosslinking the ex vivo filaments and interpretation. This is by far the most interesting topic biologically. While the limited data presented in the current submission are useful, I believe that some additional work could make the story much more compelling.

The crosslinked bands on the SDS-PAGE look very similar to the in vitro situation and this is encouraging. Unfortunately, the obtained crosslinks in this case are very few indeed. There are only two interdimer crosslinks (and both between the coil2 segments), which makes it impossible to decide on the relative location of all rod segments and, correspondingly, on the presence or absence of the various Z-folded regions in the native sample. The latter is a central question that remains unanswered at the moment.

I really wonder if these ex vivo crosslinking experiments could be repeated with the aim of obtaining more detailed data, which appears feasible (apart from the inability to use the light and heavy chains). Moreover, if the MS-based detection of crosslinks is robust then one should expect most of the in vitro intra/interdimeric crosslinks to be present in the ex vivo sample.

In addition, rotary shadowing images of the ex vivo material need to be presented, ideally with the proof that this very sample reveals the 20nm periodicity (which is assumed in the model shown in Fig. 4b).

Figures should be large enough to be legible in printed form.

Reviewer #3 (Remarks to the Author):

Makarov et al. present an interesting and feasible hypothesis that the length and flexibility of lamin A rod changes through electrostatically-driven folding of the coiled-coil segments onto each other around flexible linkers. Moreover, they postulate that the interactions between rods within the

lamin network are also mediated through folding of the disordered tails onto the coiled-coils of the neighboring rods, which offer additional flexibility. They support their hypothesis by rotary shadowing electron microscopy (EM), cross-linking mass spectrometry (CLMS) coupled with SILAC labeling to distinguish intra- from inter-molecular crosslinks and intra-dimer from inter-dimer crosslinks, crosslink-guided modeling, analytical ultracentrifugation, and site-directed mutagenesis.

Overall, I “feel” the experimental part is robust and the modeling shows at least that such folding could be accommodated. It is amazing how conformational changes can now be measured with crosslinking and what possibilities for studying homo-oligomers are now open by coupling with SILAC and other labeling techniques. However, the way the manuscript is written makes it difficult to say for me whether it is just a “feeling” or an opinion well supported by the data. It seems that this can be addressed by restructuring the manuscript, some more explanations in the main text, and more careful phrasing.

First, it is quite difficult to find the information supporting the statements in the text and to “connect the dots”. The manuscript contains a very short main text and an extensive supporting information, which includes supplementary discussion, several figures and tables, and supplementary methods. I thank the authors for trying to provide a brief and easy to read main text supported by a super exhaustive supplement. However, with such an extreme simplification of the main text, this complicates rather than simplifies the reading as I needed to constantly look into extended figures and supplementary text to even understand the main text. It often feels like an information needed to understand a statement is scattered through all over the manuscript and supporting data. Statements are often referring to multiple panels, figures and extended data at once, causing one to search through those to find the relevant information. On top of it, some figure panels are not exhaustively explained (some examples in specific comments below). Perhaps then the Results section could then be subdivided into sections and bring more explanations and detail from the supplementary discussion to the main text. The last section could then represent Discussion where Z-folding could be introduced, as the Z-folding seem to be still just a speculative model (see below).

Second, the manuscript emphasizes the Z-folding (although the abstract and the last paragraph does not). However, I think only a general flexibility or at most ensembles of various meta-stable conformations can be supported by the data, as somewhat presented but hidden in the extended data. Specifically:

- The presented modeling on its own does not seem to provide an additional evidence for Z-folding but rather provides visualization of that this hypothesis is possible under the assumption that Z-folding occurs. In fact, all other models – not parallel folds and U-turns are discarded, in Methods lines 231-233: “Of the remaining models only those with two dimeric fragments in parallel or close to parallel orientation (Z-folds) were picked for further analysis, while the rest (U-folds) were discarded as biologically non-significant”.
- Couldn’t “double U-folds” be possible and consistent with EM data where two consecutive U-turns are in close proximity separated by short coiled-coil segment? For example, in the case of

L12 and L3 separated by PH domain, the rod could bend back with U-turn on L12, follow backward with PH domain, and then bend again at L3 with U-turn again, regaining the original direction of the rod

- Since you get so many models, couldn't be just general flexibility rather than Z-folding in particular? The "stable models" obtained with Rosetta also seem to only prove that oppositely charged surfaces can in principle form interfaces but I am not convinced if this modeling prediction is accurate enough to suggest stable Z-folded conformations.
- In the above lines, what do you mean by "biologically non-significant" here? Inconsistent with EM?

Altogether, it seems that a general flexibility of the linkers, non-parallel conformations and double U-turns close to each other would also explain the data. While Z-folding is interesting and tempting, I am not sure if it is more convincing than other conformations. Perhaps then the models should be presented more as model ensembles or even localization densities (<https://www.ncbi.nlm.nih.gov/pubmed/29211988>). Z-folding could be then presented in a dedicated Discussion section as one of the possibilities.

Third, throughout the manuscript text and figures the crosslinks are always presented in categories like "short range", "long range" in Figure 2ab, "head-coil- 1A" and "tail-coil-2 "cross-links (Figure 3bc), "inter-dimer crosslinks" (Figure 3d). This is useful when discussing a particular aspect but in general quite confusing: Are these groups always disjoint? What is the complete list of the groups? Are there any crosslinks not falling into these groups? Are there any crosslinks that could not be explained by Z-folding or tetrameric interactions? Are the crosslinks on Figure 3 all crosslinks made by tails and heads or just the selected categories? Where is a figure showing that it is not just everything crosslinking to everything? Could you then add a table or figure summarizing all groups in one place and giving a global view of all crosslinks? Or indicate where to find this information (note that with such massive supplementary data it may be difficult to find such info).

Other specific comments:

- Figure 1b: can you really accurately estimate the length of the rods in the images from rotary shadowing EM? Couldn't the tails be not visible due to poor signal to noise ratio or other limitations of the method?
- Line 88: What do you mean by "unique" in "233 unique cross-links from dimer-rich bands"? Do you mean unique crosslinked residue pairs?
- Line 94: Could you explain "hendecad" – it may be not clear to readers not familiar with coiled-coils
- Line 93-94: "heptad or parallel hendecad", why hendecad is specifically mentioned to be parallel while heptad is not?

- Could you add “head” and “tail” to the domain diagram in Figure 2a?
- In lines 105-116 it is hard to appreciate the support of the models by crosslinks because the statements are very superficial “the majority of cross-links (...) supported”, “Some cross-links could not co-exist with others”
- Could you explain the meaning of I_{sc} and “stable model” in the main text?
- Fig 3b – what are the crosslinks in the dashed representation? Why is one of the crosslink green? Why the dashed crosslinks are not shown on the structure? I suppose the other crosslinks are only satisfied in alternative models, but this should be explained
- How were the models head-to-coil and tail-to-coil shown in Figure 3 and Extended Figures generated? I suppose there are some conceptual manually generated models but this could be explained
- What does “Structural projection models” mean in Fig. 3? If they were generated manually, maybe replace by “Conceptual models” If they come from ensemble generated by Rosetta modeling, replace with “Example models” (as in Figure 2c).
- Extended data Fig. 5. Panels a-c are discussed together without a clear explanation of what is a, b, and c – hard to follow
- Extended data Fig. 5d.: Line 520 speaks about “42 out of 48 input cross-links” while in the figure I can count only 14 crosslinks. The number of unique residue pairs should be used to calculate crosslink satisfaction numbers, otherwise the support looks artificially boosted (42 unique restraints is a lot!).
- The modeling of the starting fully stretched lamin A dimer based on homologous templates and with de novo fragments using CCbuilder is reasonable at this resolution of modeling. However, what is the evidence that lamin A contains PH domain as vimentin, which was used for homology modeling of the PH region?
- In Methods, line 221: “Intra- and inter-molecular cross-links were used in duplicates (i.e. from chain A to chain A and from chain B to chain B)” shouldn’t inter-molecular cross-links be from chain A to B and B to A?
- In Methods, lines 221-223 “Intra- and inter-molecular cross-links were used in duplicates (i.e. from chain A to chain A and from chain B to chain B), cross-links that happen both ways were used in tetraplets (A-A, A-B, B-A, B-B)” – this would artificially drive modeling to satisfy all possible pairs for a given crosslink while satisfaction of a single one is sufficient to explain the crosslink. In other words, this implementation implies a restraint: A-A AND A-B AND B-A AND B-B while it should be A-A OR A-B OR B-A OR B-B. The “AND” restraint may artificially lead to larger interfaces and more parallel conformations. Was the xWalk filtering also assuming “AND” restraint? This should be explained or modeling should be repeated or text should be changed to avoid proposing any specific conformation such as Z-folding
- It would be beneficial to make the models publicly available as supplementary data or through dedicated databases such as <https://pdb-dev.wwpdb.org/>. Since multiple models are

possible, they could be deposited as NMR-like model ensembles or localization densities (<https://www.ncbi.nlm.nih.gov/pubmed/29211988>).

Response to Reviewer's Comments

Many of the reviewer comments reflect the fact that the manuscript had originally been formatted for *Nature* and trying to condense the text to accommodate their size limitations made certain points confusing. We have thus addressed these comments by adding more explanation and depth to the text. Correspondingly, we have also brought into the main text most of the data from two supplemental files. We have additionally moved the panel dedicated to nucleoskeletal mechanics to new separate figure, expanding the number of figures in the main text from 4 to 7. We think these changes greatly improve clarity and readability.

A second request principally from reviewer 2 was to see if we could get more depth of *ex vivo* crosslinks by repeating this with the latest mass spectrometers. As the actual mass spectrometry data was only between 1 and 2 years old, we did not anticipate that there would be an increase in sensitivity just by re-analysing the same samples; however, we attempted two approaches that we considered might increase the coverage of cross-linked peptides. First, an extraction of the isolated rat liver nuclear envelopes with detergent would enrich for the lamina while removing many other proteins so that the lamin signal to noise ratio would increase. Thus, we prepared new nuclear envelopes from rat liver and extracted them with Triton X-100 and KCl prior to crosslinking and analysing on the mass spectrometer. Secondly, as our *in vitro* data was from human and the *ex vivo* from rat, we considered that we might increase the identification with a targeting approach after effectively training the machine with data from purified bacterially expressed rat lamin A. We did both of these and identified a few additional cross-links that support the models we presented; however, the total number of cross-links did not increase by much nor was the data or conclusions in any significant way changed. Thus, we have not changed anything in the figures for this data other than to add these cross-links.

Finally, in terms of more wide-ranging changes, it was pointed out that the term Z-fold implies more than just the overlap of the two ends of the coiled coils as it suggests a particular type of folding of the unstructured linker region. We agree with this point and have accordingly changed this usage throughout the manuscript, now using tandem stagger instead of Z-fold and anti-parallel instead of U-fold.

Reviewer #1 (Remarks to the Author):

In their manuscript „lamin A molecular compression and sliding as mechanisms behind nucleoskeleton elasticity“ Makarov et al. investigate how lamin A self-assembles into a filamentous nucleoskeleton polymer. Using mainly rotary shadowing EM and a very thorough and comprehensive analysis by crosslinking mass spectrometry together with SILAC labelling the authors can demonstrate how the intrinsically disordered head, tail and linker regions are likely connected to each other in different lamin A complexes and how this relates to the overall arrangement, assembly and mechanics of the nucleoskeletal filament.

Apart from the fact that the manuscript is very densely written and thus at times very difficult to follow, it contains a large number of thoroughly executed experiments leading to novel and -as far as the reviewer can judge - biologically relevant information, that allow the authors to propose a novel mechanism of molecular compression and sliding being the cause of lamin A higher order assembly.

Given the biological relevance of this nucleoskeletal filament in health and disease, the work by Makarov et al., certainly deserves publication in *Nature Communications*.

We thank the reviewer for their strong support of our work.

However, some points, in particularly addressing the validation of the workflow used in this study to

detect SILAC labelled crosslinks, need to be addressed before the manuscript is ready for publication.

Comments:

1. The adaption of the particular search engine used in this manuscript to detect SILAC labelled proteins should be validated using a well-established homo-dimer that is known to more easily associate and dissociate than Lamin A, preferably using all variations as input (e.g. both proteins non-labelled, both proteins SILAC labelled and labelled/ not - labelled).

At the very least it should be tested how many false positive hits the workflow is generating if no SILAC labelling is used - so using non-labelled lamin A but searching the data using the SILAC settings (e.g. allowing R10 and K8 as variable modifications).

As the reviewer uses the phrasing “At the very least” we think the reviewer likely realises that do the first suggestion would effectively constitute an entirely new study that would take additional years. Thus, we have focused on this latter suggestion. Our understanding is that the reviewer asks us to experimentally determine a false discovery rate of the particular search engine when K8 and R10 modifications are allowed as variable when employed to search label-free material. We note that *in silico* FDR calculations were already carried out in our study for each experiment, partially tackling the reviewer’s request.

We did however re-search raw data from cross-linking experiments with homo-iso-dimer mix and with homo/hetero-iso-dimer mix - each with and without K8 and R10 as variable modifications (data shown in Reviewer Only RO Table 1). We admit that allowing K8 R10 variable modification does increase the number of unique identified cross-linked residue pairs between the two experiments. However we argue that in this study this had no dramatic effect on the number of false positive cross-links identified and used to derive structural information about lamin A. Processing of the raw data for each experiment involved an appropriate FDR calculation. This was further followed by a *match_score* cut-off filtering (*match_score* is a function of quality of MS2 fragmentation spectra when compared to the theoretical spectra in the library) – allowing higher confidence in identified cross-linked spectra used for downstream analysis. And, finally and most importantly, downstream SILAC analysis was initially carried out manually in Thermo-Xcalibur and later semi-automatically with Skyline (as described in response to another request by the reviewer below). So for every filtered MS2 cross-link spectra we manually checked the quality of its pertinent MS1 spectra for the presence LL, HL, LH and HH isotopic clusters. Thus only cross-links with robustly represented LL and HH isotopic cluster or with all 4 isotopic clusters with overlapping elution peaks (matching extracted ion chromatograms XiC) were used in this study. This indeed allowed discarding several false positive cross-links and further allowed us to exclude isotopic peaks, overlapped by contaminant ions during ion peak intensity/XiC area quantification.

2 The authors should demonstrate that after gel extraction and refolding hetero-iso-dimers (dimers consisting of one heavy chain and one light chain) are indeed formed.

In the extended data figure 1a the authors show that after incubation of untagged and His-tagged lamin A the untagged lamin A elutes already in the wash fraction from the nickel column and not together with His-tagged lamin A in the eluate fraction containing 250 mM imidazole. Thus, in order to really prove that hetero-iso-dimers are indeed formed under the conditions used (e.g. gel extraction and refolding) – this experiment should be repeated with gel extracted and refolded untagged lamin A and His-tagged lamin A to show that after gel extraction and refolding untagged lamin A co-elutes with His-tagged lamin A from the nickel column.

We thank the reviewer for the suggestion of this very logical control experiment and were glad to carry this out. We have now done this experiment as shown in the new Supplementary Fig. 1a. We mixed the untagged and 6xHis-tagged lamin A, co-gel-extracted and refolded this mix and performed Ni-NTA

pulldowns with either native or gel-extracted and refolded material. We clearly observe that native tagged and untagged lamin A eluted separately; however ~50% of gel-extracted untagged lamin A was retained on the beads and co-eluted with 6xHis-tagged lamin A in 250 mM imidazole. This clearly shows that our core system works in terms of mixing the heavy and light lamin molecules in the dimers via gel-extraction and refolding. To remove any similar doubts that the potential readers of our paper may have, we included this result into the Supplementary Fig. 1a and described its logic in the main text.

Minor Points:

1 For the definition of the Inter-chain / inter-molecular / inter-dimeric ratios in the supplementary discussion the authors mention: “likewise, in the second experiment (...) a crosslink occurring only between similarly labelled pairs of peptides would be a cross-link happening strictly within a single molecule of lamin A”.

Could in such an experimental setting a crosslink with similarly labelled peptides (e.g. L/L or H/H) not also originate from different lamin A molecules (either different molecules of a dimer or different dimers)? Maybe the authors could comment on this.

We apologise for the lack of clarity. The active word for our intention was “only”, the idea being that the same cross-link would generally be recovered as LL, HH, LH, and HL and so if we only obtained the LL and HH without HL and LH we would expect that the cross-link was intra-molecular. For all of the analysed MS2 spectra for a particular pair of cross-linked residues we never relied only on actual MS2 identification, but monitored the presence of all 4 – LL, HL, LH and HH - isotopic clusters in the pertinent MS1 and XiC.

2 For reasons of completeness and if easily done, it might be nice if the elution peak areas of the 4 peptide-pair precursor ion clusters used for the manual quantification in Xcalibur? (The 'LFQ' values to calculate the ratios) could also be provided (and not only the ratios).

We are happy, to at least partially fulfil this request. At the time of the data processing for the original experiments and until very recently there was no tool that would allow a completely automatic recording of elution peak areas in SILAC CLMS experiments where each cross-link can happen not only between a pair of light peptides or between a pair of heavy peptides (LL and HH), but also between light and heavy peptides (HL and LH). As it stands, most platforms like MaxQuant are incapable of recording LFQ values and performing ratio calculations in such an experiment. This is large part due to the fact that in any cross-linking experiment cross-linked peptides would be of very low abundance and represent a minority of protein material after tryptic digestion (most peptides would be just linear uncross-linked). Thus in most cases, even if cross-links physically happened between all possible combination of peptides (LL, HL, LH and HH), only one or two of these would be picked up in MS1 for a consequent MS2 fragmentation in a narrow elution peak. Therefore remaining “un-identified” peaks would have to be tracked manually to extract peak intensity/XiC data. During the initial data processing this was thus done manually in Thermo-Xcalibur, which, unfortunately, lacked the sufficiently convenient interface functionality to record extracted ion chromatogram (XiC) areas during the manual inspection of 16,476 peaks carried out in this study (three isotopic peaks for each of the 4 – LL, HL, LH and HH ion clusters multiplied by the number of cross-linked spectra). Therefore, instead of the XiC peak area, we had to rely on the intensity values of each isotopic peak as averaged across matching elution peaks. As these were to be used for ratio calculations, they were recorded to the second decimal of their respective order of magnitude. We will be happy to provide the complete record of these values after sufficient clean-up for the final publication.

However recently and through the joined effort of the Rappsilber group and Skyline development team Skyline software became capable of semi-automatic tracking LL, HL LH and HH peaks in the raw data files even in the absence of all 4 MS2 identifications (Chen, Z. A. & Rappsilber, J. *Quantitative cross-linking/mass spectrometry to elucidate structural changes in proteins and their complexes. Nature protocols* **14**, 171-201, doi:10.1038/s41596-018-0089-3 (2019)). Unfortunately, this software still requires frequent manual correction of the elution peak windows and filtering out contaminant ions. Therefore we were glad to employ Skyline for a set of 162 spectra as a proof of concept and to assess whether our original calculations were correct. We are happy to report that the average difference between manual average peak intensity ratios (HL+LH)/(LL+HL+LH+HH) and the same ratios calculated using Skyline XiC areas was less than 2% and individual spectra errors were within +/-5% for 95% of measurements, excluding 6 spectra with obvious contamination problems (i.e. one or more isotopic peaks overlaid by contaminant ions as evidenced by differing XiC and poor isotope ratio fit). This data is now included into the Supplementary table 1 in the tabs “Skyline XiC H/hIDm X”, “Skyline XiC HIDmX” and “Skyline vs Xcalibur”.

3 What is meant by „Z-folding“ of the linker? (EXPLAIN)

As noted above, we have re-evaluated our use of the term Z-folding and decided that it could be interpreted to mean more structurally than the data supports. We had initially used this term because cross-links identified between the flanking coiled coils and the linker connecting them would indicate that the linker mediates interactions between the coiled coils that instead of being in sequence are staggered. However, a "Z" could also be interpreted to have the coiled coils fully on top of one another instead of staggered and it implies a particularly ordered set of interactions of the linker between them. Our modelling would rather suggest that the structure is more like if one had a set of Japanese nunchaku or nunchucks as they are more commonly referred to, but instead of holding them like normal (which would effectively be the U-fold), one were to first hold them linearly fully stretched out and then bring one partly up over the other with the chain being able to adopt many conformations in interacting with both rods. While on the one hand this is a bit complex to convey without metaphors, due to our inability to think of a perfect metaphor, we have decided to revert our language to simply calling it a tandem stagger where the linker can interact with both coiled-coil termini. We have changed this throughout the text as well as changed the "U-fold" to "anti-parallel fold".

4 In line 91 in the text it is stated that 'EDC cross-linking captured rod shortening (Fig. 1 c: X, 1-3) ...' it seems the text should refer to Fig 1f?

We have fused some panels in Figure 1 so that f is now g. However this was supposed to refer to the “lengths distribution diagram in c” and accordingly we have changed this to "*Rod measurements of the dimer-rich cross-linked material further revealed that cross-linked dimers, while widely distributed for lengths, were significantly shorter than those in the uncross-linked NaPi control (Fig. 1c 1-3)*".

5 Would it be possible to strengthen the point made from the gel in figure 1f by directly measuring the length of the dimers by rotary shadowing EM to directly demonstrate that the different crosslink bands 1, 2 and 3 contain dimer fractions of a preferred length? E.g. dimers with a length of 40-44 nm are significantly enriched in band 1, whereas band 3 contains more dimers with a length of 48-50 nm.

The point we were making from the gel was that the higher molecular weight bands contained tetramers. We made no such conclusions regarding dimer rod lengths because there was no statistical significance between the distributions for bands 1-3 (R stats give Dunn-Holms 10^{-1}) non-significant difference, Fig.

1c). We only noted that all three of these bands were statistically significant in differences with the NaPi condition. We postulate that the migration in the gels for the different bands has more to do with head and tail folding than rod length; however, as there is insufficient data to unequivocally answer this question we did not state this in the manuscript.

6 What is the definition of “short range” and “long range” crosslinks?

"Short-range cross-links" is the term we used when the length between cross-linked amino acid residues is within the range possible for the EDC cross-linker while satisfying pre-existing structural predictions for coiled-coils. "Long-range cross-links" are those that do not and so indicate a requirement for a new structural model. We acknowledge that both usages are somewhat confusing because the cross-links are neither short- nor long-range, but rather the structural predictions either fit or don't fit the limitations of the cross-linking chemistry. As another reviewer also had issue with the usage of these two terms and our extensive categorisation of cross-links we opted to altogether avoid these terms. Instead in the main text when talking about cross-links within the coiled coil segment termini we gave a more detailed description: *“Importantly, as all of these cross-links either already fit the predicted linear coiled-coil structure or can be satisfied via axial chain rotation and small interruptions in individual chain α -helical structure to bring respective residue side chains in contact for EDC cross-linking, none of these cross-links can directly account for rod shortening. Interestingly, of the remaining 19 cross-links within individual coiled-coil segments, an additional 16 cross-links were between residues in the coiled-coil segment termini but too distal for EDC cross-linking in the predicted coiled coil structure even with small structural changes and thus these imply more severe irregularities in these termini (Fig. 4a, red cross-links).”*

Figure 4 (was Figure 2) was changed to match.

7 With the exception of Rosetta, the exact software versions of the used software programs are missing and should be added and the settings used for modelling should be explained.

This has been addressed.

8 It would also be highly advisable to re-run at least some of the Rosetta runs using only a subset of the crosslinks as input to assess the robustness of the obtained models.

This has been attempted for coil 1A and coil 1B adjacent termini fragments resulting in a slightly smaller set of stable models (156 satisfying 16 out of 24 input cross-links as opposed to 220 satisfying 18 out of 24 in the data presented in the manuscript). We note that, as we were looking for possible stable (as defined by the difference of energies of the complex and partners in isolation) interfaces between the pairs of docked coiled-coil fragments that satisfy cross-links we do not necessarily require an exhaustive set of models for each of the pair of docked coiled-coil fragments to demonstrate this point. The number of generated models for coil 1A/1B overlap and PH/coil2 overlap we have already points towards high redundancy in the overlap interaction. However, we also find that docking in Rosetta using cross-linking constraints as an input in search of possible stable interfaces satisfying these cross-links was not necessarily the best approach in our specific case: as mentioned in the main text some of the cross-links could not be satisfied in combination with others in the same model, but Rosetta was attempting to satisfy as many cross-links as possible at the expense of individual stable models. An alternative approach was to dock coiled-coil fragments without constraints and then filter models satisfying cross-links afterwards. This has been attempted and yielded a staggering number of models for coil 1A/1B overlap, among which

over 1,500 stable models satisfied cross-links. This has been now reflected in the text: “*The true extent of this redundancy became further apparent in additional in silico docking experiment attempted for adjacent termini of coils 1A and coil 1B using Rosetta without cross-linking constraints. This generated a much larger set of relatively parallel stable tandem stagger models, of which over 1,500 satisfied one or more of each of the 6 input cross-links in any of their intra- and inter-chain variations*”. Note: 6 identified cross-links were both possible as intra- and inter-chain according to our quantification, giving rise to the total of 24 variations (previously talked about as unique cross-links).

9 It might be a good idea to deposit the basic lamin A model, the Rosetta output (as well as the refined output) and the code snippets inside a publicly available git repository. This is especially true for the Modeller code since Python code is sensitive to white spaces and might be changed by the print formatting.

As were running additional modeling as suggested above and also by reviewer 3 and with some code modifications, this is only just happening. However, the data will all be publicly available upon publication from Edinburgh Datashare.

10 According to the supplementary information crosslinks were categorized and validated manually as well as via Xwalk. It would be nice to add a table showing the results of this validation as well as the calculated crosslink distances.

This has been addressed for cross-links within coiled coil rod segments and cross-links across L1 as it was modelled as α -helical in the prediction-based model. Supplementary Table 2 contains C α -C α distances for cross-linked residue pairs along the rod axis and these values are now supplemented with surface distances (SAS) between cross-linked residue C β atoms. Distances above 30 Å were not calculated as residues with C β atoms further than 12-14 Å would be outside of the EDC physical range.

Reviewer #2 (Remarks to the Author):

The manuscript reports on the use of chemical crosslinking towards elucidating the structure of nuclear lamins. In vitro assembled lamins have been studied first, revealing the structure of the dimer as the dominant solution species. In addition (and somewhat surprising since the conditions used are those supporting soluble dimers rather than further assembly), data on the head-to-tail contact of the dimers have been obtained. This contact reveals critical interactions of the positively charged head and tail regions with the negatively charged rod ends, in line with existing predictions. In addition, some limited crosslinking data have been collected on ex-vivo lamina.

Chemical crosslinking has to my knowledge not been reported for lamins before, and the wealth of obtained crosslinking data (generally sound, with some reservations mentioned below) is the main strength of the manuscript. The conclusions about the possible ‘Z-folding’ of selected parts of the dimeric rod are novel and generally appearing worthy, while the biological significance still largely remains unclear, beyond some speculations provided.

The manuscript has apparently been initially written for a different journal and contains, first, an extremely condensed main text and main figures, followed by a set of supplementary figures (with lengthy legends, discouraged in Nature Comms), methods and supplementary discussion on crosslinking approach. I found this layout extremely cumbersome. The main text includes many conclusions that

sound totally unconvincing unless the reader works own and hard way through all additional sections. Getting a clue on the crosslinking approach used is impossible without reading the supplementary figure legends and also the (very useful) supplementary discussion. A figure encompassing all crosslinks within the dimer is nowhere to find; to get an idea of just the intra-rod crosslinks, one has to examine Fig. 2b, then extended data Fig. 2a and also extended data Fig. 2f!!

The reason we did not previously have a figure showing all the cross-links together is because by spending many years on this project and running many samples we have so many cross-links that individual ones cannot be readily distinguished in such a figure. However, we reported all cross-links obtained. We have now addressed this also by merging all the cross-links in the rod into panel 3a and we show all the cross-links within the rod and all cross-links with residues outside the rod in panels 3b and 3c and summarised respectively in Supplementary Tables 2 and 4.

As for the other point, indeed the reviewer is correct in their assessment that this was originally written for *Nature* and many of the reviewer's comments reflect our inability to show all data in the main text with the size constraints of that journal. While these issues in the current instance reflect the transference of manuscripts within the *Nature* family, we have in this revision accordingly expanded the text that should address all the issues that made the original text more difficult to follow.

The very beginning of the main text is quite unfortunate. The authors present some data such as rotary shadowing and UC towards a variability of the dimer length and its shortening upon crosslinking. These data are neither too convincing nor telling. Next, while the 20nm periodicity of the lamin filaments was seen in recent cryo-EM work on native lamina, discussing the 'in vitro' 25nm periodicity (seen in paracrystals, possibly?) is not too relevant.

The 25 nm periodicity is not based only on paracrystals and the 52 nm rod of lamins remains in all the textbooks. In fact, in the *Nature* paper from Ohad Medalia's group they suggested a greater overlap between molecules rather than rod shortening to explain the periodicity they measured. Thus, this finding, which is actually very strong because two independent methods of analytical ultracentrifugation and rod measurements support it, is actually extremely relevant. Moreover, apart from the extreme relevance to the whole intermediate filament field, this finding is relevant in the bigger context in that it shows that historical views of measuring just the longest one can find and attributing that to an entire population is wrong and that investigation of other molecules similarly may reveal variations that have thus far been ignored. In fact, these findings parallel the excitement around how single cell measurements have revealed considerable variation that is biologically relevant.

Most likely the interesting crosslinking data and the Z-folding models have been the starting point of all these questions, and they should be reported first. The in vitro variability of the dimer (a topic with limited applicability to the native situation anyway) could be discussed thereafter as providing some (limited) support to the Z-folding possibility.

Overall the current manuscript is too lengthy and should be best split in two separate manuscripts IMO.

We believe that the issues that caused the reviewer to recommend such significant changes in the manuscript organisation are due to our inability to communicate all points effectively for the very short format of *Nature*. In revising it for *Nature Communications* we have greatly expanded the text to make it more clear and anticipate that as neither of the other reviewers saw the need for such significant changes that Reviewer 2 will find that the many changes we have made have sufficiently improved the manuscript that such drastic changes in organisation are not needed.

1. The first manuscript could focus on in vitro crosslinking (the strongest part of this work as mentioned) and dimer/ H2T tetramer modelling. Fully detailed account of the original crosslinking and modelling approaches, presented logically without the need to hunt through endless supplements, would be an interesting reading.

I have a bit of concern about the validity of deciding on the type of crosslink (intrachain, intradimer or interdimer) from the heights of MS peaks. It is known that the latter are often not representative of the relative abundance of different peptides, which may 'fly differently' in MS.

It is certainly true that quantification of relative abundances of two different peptides (or two different pairs of cross-linked peptides) based exclusively on peak intensities and without additional normalisation would not be accurate. However in our case such comparative quantification was done every time only for peptides of the very same sequence and cross-linked via the same pair of residues with the only difference being the degree of isotopic labelling in the pair: pair of light peptides, pair of heavy peptides and mixed LH and HL peptide pairs. Isotopic labelling is widely used in classical SILAC experiments and is not known to affect "flying ability" of linear or cross-linked peptides.

As for the "peak heights" – we have now compared quantification methods using manually derived average across the elution profile peak intensities in Xcalibur and semi-automatically recorded extracted ion chromatograms areas in Skyline (Chen, Z. A. & Rappsilber, J. Quantitative cross-linking/mass spectrometry to elucidate structural changes in proteins and their complexes. *Nature protocols* **14**, 171-201, doi:10.1038/s41596-018-0089-3 (2019)). This was done for a set of 162 spectra and we are happy to report that, with the exception of several individual cases where identification of one or more isotopic clusters in MS1 (LL, HL, LH or HH) was complicated by contaminant ions, the manual and semi-automatic quantification yields very similar results. Differences between manual average peak intensity ratios (HL+LH)/(LL+HL+LH+HH) and the same ratios calculated using Skyline XiC areas were less than 2% and individual spectra errors were within +/-5% for 95% of measurements. This data is now included into the Supplementary table 1 in the tabs "Skyline XiC H/hIDm X" "Skyline XiC HIDmX" and "Skyline vs Xcalibur".

In addition, I wonder if an independent proof of the resulting structural models can be obtained using a very different chemical crosslinking strategy, such as with the classical BS3 for instance?

Although we do not use any of the data in the paper in order to try to make it easier to follow, we actually did initially test other cross-linkers including BS3 in setting up this study and obtained cross-links also consistent with these models. However, most well-established cross-linkers such as BS3 feature a long cross-linker arm that in turn greatly reduces the spatial resolution of cross-linking data. EDC is a zero-length cross-linker that requires immediate proximity of the residue side chains to work. As the reviewer has noted elsewhere that there is already too much data that it is hard to sift through, we think we made the correct decision on this and so have not increased the complexity of this study by adding more.

Nearly all crosslinks used to produce the Z-folded models (Fig. 2b) appear to be both intrachain and intradimeric. Is there a logical reason for this?

It is clearly evident from Supplementary Tables 2 and 3 that this is not the case as the majority of those cross-links were both intra and inter-chain, which further attests to the flexibility of the linkers.

2. Second manuscript could focus on crosslinking the ex vivo filaments and interpretation. This is by far the most interesting topic biologically. While the limited data presented in the current submission are

useful, I believe that some additional work could make the story much more compelling.

The crosslinked bands on the SDS-PAGE look very similar to the *in vitro* situation and this is encouraging. Unfortunately, the obtained crosslinks in this case are very few indeed. There are only two interdimer crosslinks (and both between the coil2 segments), which makes it impossible to decide on the relative location of all rod segments and, correspondingly, on the presence or absence of the various Z-folded regions in the native sample. The latter is a central question that remains unanswered at the moment.

We fear the reviewer may not realise how difficult it may be to obtain cross-linked species in this type of study. The number of cross-links identified here reflects pretty much the best that can be achieved with this technology for this kind of protein. In recent years great strides were made towards improvement of cross-linked peptide recovery from complex *in vivo* samples, however only in very few cases can more than a handful of cross-links be obtained. One of the main approaches to reach cross-link recovery levels nearing those attainable *in vitro* is to utilise affinity purification of target protein complexes (Shi, Y. *et al.* *A strategy for dissecting the architectures of native macromolecular assemblies. Nature methods* **12**, 1135-1138, doi:10.1038/nmeth.3617 (2015)). This was utilised to a great success to dissect the structure of the nuclear pore (Kim, S. J. *et al.* *Integrative structure and functional anatomy of a nuclear pore complex. Nature* **555**, 475-482, doi:10.1038/nature26003 (2018)). However this approach is poorly applicable in our case as, unlike NPC proteins, lamins are assembled into a nuclear-sized polymeric structure underlying the inner nuclear membrane and interacting with hundreds of different proteins (Dittmer, T. A. *et al.* *Systematic identification of pathological lamin A interactors. Mol Biol Cell* **25**, 1493-1510, doi:10.1091/mbc.E14-02-0733 (2014)), including a myriad of nuclear envelope proteins, NPC components and chromatin. Chemical cross-linking of such vast polymeric structure in presence of its multiple partner proteins would render conventional pull-down techniques useless.

Alternative approaches rapidly developing in recent years are not yet widely employed and rely on MS-cleavable chemical cross-linkers such as DSSO and a currently unavailable to us MSⁿ-based (as opposed to MS2) CLMS analysis (Liu, F., Rijkers, D. T., Post, H. & Heck, A. J. *Proteome-wide profiling of protein assemblies by cross-linking mass spectrometry. Nature methods* **12**, 1179-1184, doi:10.1038/nmeth.3603 (2015) and Yu, C. & Huang, L. *Cross-Linking Mass Spectrometry: An Emerging Technology for Interactomics and Structural Biology. Analytical chemistry* **90**, 144-165, doi:10.1021/acs.analchem.7b04431 (2018) for review). To our knowledge Liu *et al.* have recovered the most cross-links within various proteins than any other single study not relying on affinity purification of target protein complexes; and certainly the most lamin A cross-links – 21 cross-links with DSSO. This used an NHS-ester-based homobifunctional cross-linker that features a 10.3 Å spacer arm and primarily targets lysine, and depending on cross-linking conditions, serine, threonine and tyrosine residues.

After additional experiments we are now confident in 15 cross-links in lamin A that all indicate immediate residue proximities: EDC is a self-excluding zero-length cross-linker that targets immediately proximal residues and actual electrostatic and polar interactions (D/E-K/T/S/Y residue pairs). While the number of cross-links we recovered has not greatly improved, we are now finding additional and more robust evidence for the L1 tandem stagger (previously Z-folding) and the overlap of coils 1A and 1B, new evidence of interactions of the head domain with the coil 1A as well as that of L12 tandem stagger folding and overlap between coils 1B and the PH region of coil 2. These have been included into the new Figure 6 (was Figure 4) and added to the Supplementary Table 5.

I really wonder if these *ex vivo* crosslinking experiments could be repeated with the aim of obtaining more detailed data, which appears feasible (apart from the inability to use the light and heavy chains).

Moreover, if the MS-based detection of crosslinks is robust then one should expect most of the in vitro intra/interdimeric crosslinks to be present in the ex vivo sample.

As described above, there are considerable difficulties in cross-link recovery from complex organelles. Nonetheless, we considered the possibility that due to the considerable stability of the lamin polymer (e.g. resistance to extraction with 1 M salt or 2% detergent), biochemical extraction might enrich for lamins sufficiently to increase the recovery of cross-linked species. We would note that this approach would have the potential to change some of the cross-links in the network because the extracted proteins might be supporting maintenance of a particular conformation. We prepared more rat liver nuclear envelopes and extracted with 1% Triton X-100 and 400 mM KCL in a procedure similar (though milder) to one described before by N. Dwyer and G Blobel (*Dwyer, N. & Blobel, G. A modified procedure for the isolation of a pore complex-lamina fraction from rat liver nuclei. J Cell Biol* **70**, 581-591 (1976)). Cross-linking of such “stripped” nuclear envelopes and consequent selective solubilisation of lamina material in 6 M urea however still yielded a mixture of 781 proteins (3 or more peptides). Lamin A sequence coverage stood at 73.4%, however, despite being the third most abundant after histones H3 and H4, it only constituted 2.85% of total protein material by MaxQuant iBAC (4.96% of top 30 proteins) (reviewer only RO Table 2). From these experiments we identified several additional cross-links including: a head-coil-1A cross-link, S12-E33; a classical heptad e-d interaction cross-link; E31-E32; E65-K97, E68-K97 all supporting coil 1A/1B tandem stagger; D192-S239 supporting coil 1B/PH tandem stagger; and E194-T199. All of these have been added to the Supplementary Table SI 5 and into the Figure 6 (was Figure 4). After thorough analysis we additionally de-validated and discarded cross-links E279-S282, E290-K542 and D375-Y359, bringing the total number of *ex vivo* captured cross-links we are confident in to 15.

In addition, rotary shadowing images of the ex vivo material need to be presented, ideally with the proof that this very sample reveals the 20nm periodicity (which is assumed in the model shown in Fig. 4b).

The *ex vivo* material is assembled and according to estimates from protein gels and the mass spectrometer data the lamins represent less than 5% of the material in the ex vivo lamina fraction. The reason that the Ohad Medalia paper was a Nature paper was because it was an entire study just to try to visualise the intact lamin polymer and it would be impossible to visualise it with rotary shadowing that would just coat the many proteins assembled together and lose all resolution.

Figures should be large enough to be legible in printed form.

We have improved this.

Reviewer #3 (Remarks to the Author):

Makarov et al. present an interesting and feasible hypothesis that the length and flexibility of lamin A rod changes through electrostatically-driven folding of the coiled-coil segments onto each other around flexible linkers. Moreover, they postulate that the interactions between rods within the lamin network are also mediated through folding of the disordered tails onto the coiled-coils of the neighboring rods, which offer additional flexibility. They support their hypothesis by rotary shadowing electron microscopy (EM), cross-linking mass spectrometry (CLMS) coupled with SILAC labeling to distinguish intra- from inter-molecular crosslinks and intra-dimer from inter-dimer crosslinks, crosslink-guided modeling, analytical ultracentrifugation, and site-directed mutagenesis.

Overall, I “feel” the experimental part is robust and the modeling shows at least that such folding could be accommodated. It is amazing how conformational changes can now be measured with crosslinking and what possibilities for studying homo-oligomers are now open by coupling with SILAC and other labeling techniques. However, the way the manuscript is written makes it difficult to say for me whether it is just a “feeling” or an opinion well supported by the data. It seems that this can be addressed by restructuring the manuscript, some more explanations in the main text, and more careful phrasing.

First, it is quite difficult to find the information supporting the statements in the text and to “connect the dots”. The manuscript contains a very short main text and an extensive supporting information, which includes supplementary discussion, several figures and tables, and supplementary methods. I thank the authors for trying to provide a brief and easy to read main text supported by a super exhaustive supplement. However, with such an extreme simplification of the main text, this complicates rather than simplifies the reading as I needed to constantly look into extended figures and supplementary text to even understand the main text. It often feels like an information needed to understand a statement is scattered through all over the manuscript and supporting data. Statements are often referring to multiple panels, figures and extended data at once, causing one to search through those to find the relevant information. On top of it, some figure panels are not exhaustively explained (some examples in specific comments below). Perhaps then the Results section could then be subdivided into sections and bring more explanations and detail from the supplementary discussion to the main text. The last section could then represent Discussion where Z-folding could be introduced, as the Z-folding seem to be still just a speculative model (see below).

We thank the reviewer for these suggestions. The manuscript was originally condensed to fit the Nature format and as such we fully agree that there was insufficient space to elaborate these points. We have now subdivided the text into sections, generally expanded the text, moved some of the supplemental data into the main text as described in the overview paragraphs. This makes it so that the reader has to access much less the supplemental data and so makes it much easier to follow. As far as the Z-folding goes, we have changed the description of this to a tandem stagger, which we think better elaborates the point and addresses reviewer criticisms of the term as the Z implies the parts of the model that the reviewer correctly notes are more speculative. The data reasonably clearly show a tandem stagger, but the linker interactions are more varied and we cannot determine which interactions occur together in a particular conformation.

Second, the manuscript emphasizes the Z-folding (although the abstract and the last paragraph does not). However, I think only a general flexibility or at most ensembles of various meta-stable conformations can be supported by the data, as somewhat presented but hidden in the extended data. Specifically:

- The presented modeling on its own does not seem to provide an additional evidence for Z-folding but rather provides visualization of that this hypothesis is possible under the assumption that Z-folding occurs. In fact, all other models – not parallel folds and U-turns are discarded, in Methods lines 231-233:

“Of the remaining models only those with two dimeric fragments in parallel or close to parallel orientation (Z-folds) were picked for further analysis, while the rest (U-folds) were discarded as biologically non-significant”.

- Couldn't “double U-folds” be possible and consistent with EM data where two consecutive U-turns are in close proximity separated by short coiled-coil segment? For example, in the case of L12 and L3 separated by PH domain, the rod could bend back with U-turn on L12, follow backward with PH domain, and then bend again at L3 with U-turn again, regaining the original direction of the rod

We agree with the reviewer that “Z-fold”, despite its being a good metaphor for the tandem stagger that the data best supports, implies actually more than we intended in that it suggests a particular conformation of the linker as well. The data, while supporting that the linker can interact with both ends of the flanking coiled coils, cannot distinguish which of these cross-links recovered occurred together in the same molecule/dimer or whether these interactions were necessary for the formation of the stagger in the first place. Therefore, we have removed the term Z-fold and replaced it with tandem stagger, which is directly supported by cross-links between the two coiled coils and our modelling data.

As far as double U-folds go, while we got cross-links supporting the interactions of adjacent coiled coils, we did not get cross-links supporting more distal interactions for example between coil 1A and the hendecad at the start of coil 2. While this does not completely rule out the possibility, the disparate size of the coiled coils would argue that we would have obtained other “overlength” cross-links that could not be explained with existing models between the adjacent coils other than just at the ends where we obtained these cross-links.

- Since you get so many models, couldn't be just general flexibility rather than Z-folding in particular? The “stable models” obtained with Rosetta also seem to only prove that oppositely charged surfaces can in principle form interfaces but I am not convinced if this modeling prediction is accurate enough to suggest stable Z-folded conformations.

The reviewer is exactly correct in that we don't view any of the linker, or tandem stagger as a completely stable or absolute: rather we think that each linker is capable of a wide range of electrostatic interactions and the amount of coil overlap from the stagger can also be quite variable, accounting for the observed variation in rod length. Moreover, since our data suggest that this is largely driven by electrostatic interactions we might expect that there could be some folds like the suggested double U-fold that might occur too infrequently for us to have identified them, but that may not be able to integrate into the polymer because they don't ‘fit’. Whether constraints in the intact polymer might favour one stagger overlap from another is well beyond both this work and current science to investigate, but at least based on the cross-links identified a wide range of conformations is possible. We feel that we did not convey this adequately to get this comment and so have tried to better emphasise the flexibility in the system in the revision.

- In the above lines, what do you mean by “biologically non-significant” here? Inconsistent with EM? Altogether, it seems that a general flexibility of the linkers, non-parallel conformations and double U-turns close to each other would also explain the data. While Z-folding is interesting and tempting, I am not sure if it is more convincing than other conformations. Perhaps then the models should be presented more as model ensembles or even localization densities (<https://www.ncbi.nlm.nih.gov/pubmed/29211988>). Z-folding could be then presented in a dedicated Discussion section as one of the possibilities.

As noted above, we have removed the specific “Z-folding” usage from the manuscript and further emphasised the flexibility in the tandem stagger. We have also shown other models in Supplemental Figs. 3-4 and acknowledged the possibility of other models. However, we do not think that these other

forms would be able to pack efficiently into a reasonably stable polymer as FRAP, FLIP and much other data have demonstrated the lamin polymer to be.

Regarding the presentation of the models and ensembles or localisation densities – we believe that Supplementary Figures 3-4 adequately convey the fact that within the selected and refined pool of tandem stagger models for coils 1A/1B overlaps and PH/coil 2 overlaps respective coiled-coil fragments exhibit the general tandem stagger with a wide range of flexibility in the individual interactions. It is our understanding that the ensemble model presentation is more useful when a docking experiment yields a multitude of models distributed between several tight and distinct clusters (i.e. globular protein A docks to one side or to the other side of a bigger globular protein B) where it can be used to compare these two clusters. Thus employing such presentation here would be somewhat redundant, of not misleading.

Third, throughout the manuscript text and figures the crosslinks are always presented in categories like “short range”, “long range” in Figure 2ab, “head-coil- 1A” and “tail-coil-2” “cross-links (Figure 3bc), “inter-dimer crosslinks” (Figure 3d). This is useful when discussing a particular aspect but in general quite confusing: Are these groups always disjoint? What is the complete list of the groups? Are there any crosslinks not falling into these groups? Are there any crosslinks that could not be explained by Z-folding or tetrameric interactions? Are the crosslinks on Figure 3 all crosslinks made by tails and heads or just the selected categories? Where is a figure showing that it is not just everything crosslinking to everything? Could you then add a table or figure summarizing all groups in one place and giving a global view of all crosslinks? Or indicate where to find this information (note that with such massive supplementary data it may be difficult to find such info).

We have removed the use of short range and long range because we feel in retrospect that this was misleading since all the cross-links use the same chemistry and have the same physical constraints. These terms were used previously to reflect whether the chemical constraints from the EDC cross-linker could fit with the pre-existing models and so we have tried to describe this better in the text instead of using these potentially misleading terms. We agree that there were too many categories and this has streamlined this to hopefully partly address this reviewer comment. We did keep the segregation of cross-links by region in the molecule as this helps orient the reader to where they occurred and also some other categories; however, we have tried to make this clearer in presentation by starting with a new panel in the new Figure 3 that shows every single dimeric cross-link we obtained all in one figure and then have added to the text explaining our logic in extracting different subsets to discuss for different issues such as those supporting the long predicted coiled-coil geometries from those previously called long range and for which we now simply indicate that they do not fit pre-existing predictions and so require the need for new models. As such all the cross-links are summarised in Supplementary Table 1 (all cross-links in all bands), Supplementary Table 2 (all rod cross-links in dimeric bands with annotations) and Supplementary Table 4 (all other dimeric cross-links with annotations).

Other specific comments:

- Figure 1b: can you really accurately estimate the length of the rods in the images from rotary shadowing EM? Couldn't the tails be not visible due to poor signal to noise ratio or other limitations of the method?

We only measured from those dimers where we could clearly distinguish globular tails. Although one could argue that this itself might have biased the results, the greater probability is that measuring less clearly defined dimers would have been more likely to count the globular tails as part of the rod and so push the rod measurements into the longer direction rather than the shorter direction that we observed. We nonetheless agree with the reviewer that measurement of rotary shadowed images has the potential to be influenced by the bias of the measurer; however, the fact that the independent and without bias measurements by analytical ultracentrifugation concur indicates that we were in fact able to accurately

estimate the length of the rods from the images. The general standard in science is that when two completely different experimental approaches achieve the same result it generally means that the measurement is accurate and this is what was done in this case.

- Line 88: What do you mean by “unique” in “233 unique cross-links from dimer-rich bands”? Do you mean unique crosslinked residue pairs?

Yes. In considering this reviewer point we initially tried to state “cross-linked residue pairs” in every usage and realised that this was quite arduous on the reader. Therefore, we have now defined early on that “cross-links” always means “cross-linked residue pairs” by changing the text to: *“A total of 233 unique cross-linked residue pairs (subsequently referred to as just “cross-links”) from dimer-rich bands 1-3 and 143 cross-links from tetrameric band 4 were identified. Fittingly, SILAC quantification revealed that 229 out of these 233 unique cross-links from dimer-rich bands indeed occurred in dimers with varying frequency while 66 out of 143 cross-links from the tetrameric band were inter-dimeric (Fig. 2h)”*

- Line 94: Could you explain “hendecad” – it may be not clear to readers not familiar with coiled-coils

We have added *“A heptad is a 7 amino-acid motif that drives dimerisation of two alpha helices into a stable twisted superhelix compared to a hendecad which is an 11 amino acid motif that forms a less stable parallel bundle of alpha helices. Most of the lamin A coiled-coil rod is predicted to feature heptad structure, but at the end of coil 1B and beginning of coil 2 a hendecad structure was predicted to form”*.

- Line 93-94: “heptad or parallel hendecad”, why hendecad is specifically mentioned to be parallel while heptad is not?

There has always been an historical assumption that intermediate filament coiled coils are parallel so we did not think to specify it. However, in the text hendecad was stated as parallel to indicate a lesser degree of intertwining of alpha helices that affects the stability of the helical dimer and indeed we obtained cross-links indicating coiled-coil irregularities in the hendecad region such as the C-terminus of coil 1B and PH region. However, as we used the term “predicted” here and in fact the predictions have all to our knowledge favoured a parallel structure we have changed the phrasing to *“Intermediate filament rods are thought to consist of multiple α -helical segments — coils-1A, -1B and -2 — that dimerise into parallel heptad or hendecad coiled-coils²²⁻²⁵ accounting for ~48 nm of rod length by prediction.”*

- Could you add “head” and “tail” to the domain diagram in Figure 2a?

This has been done, though the figure is now 3a.

- In lines 105-116 it is hard to appreciate the support of the models by crosslinks because the statements are very superficial “the majority of cross-links (...) supported”, “Some cross-links could not co-exist with others”

This was due to the word restrictions in previously formatting this work for *Nature*. We have now given specific examples in the text in support of the models; however, at the same time, in accordance to other comments of the reviewer, we have better explained the limitations on the support for any individual model.

- Could you explain the meaning of I_sc and “stable model” in the main text?

We thank the reviewer for pointing this out as we did not know the term prior to collaborating with our expert Rosetta modeler and so likely many readers also would not. The I_sc (Interface_score) is a measure of interface stability in Rosetta calculated as a difference of total energy of the complex minus energies of the partners in isolation and we have now added to the text “*Some cross-links could obviously not co-exist with others within a single tandem stagger model (Supplementary Fig. 3a), but >250 tandem stagger models with stable inter-coiled-coil interfaces satisfying overlapping subsets of cross-links were found as determined by Rosetta I_sc values < -5.0 (Rosetta’s Interface score calculated as a difference between the total energy of the complex and the sum of total energies of each partner in isolation; as detailed in RosettaDock application documentation <https://www.rosettacommons.org/docs>)*”.

- Fig 3b – what are the crosslinks in the dashed representation? Why is one of the crosslink green? Why the dashed crosslinks are not shown on the structure? I suppose the other crosslinks are only satisfied in alternative models, but this should be explained

Thank you for pointing out this oversight. The green cross-links in panels b and c in this figure represent the ones shown on the conceptual models underneath. The reviewer is further correct in that dashed cross-links are likely to be satisfied in other structures. We have further added to the figure legend that both conceptual models in panels b and c are just example models satisfying some of the cross-links observed. Alternative models are presented in Supplementary Figure 6. We have now explained all these points in the figure legend and additionally solidified dashed lines.

- How were the models head-to-coil and tail-to-coil shown in Figure 3 and Extended Figures generated? I suppose there are some conceptual manually generated models but this could be explained

Yes, these were generated manually to assess whether unstructured head and tail regions sport sufficient length to satisfy different cross-link sets and as such reflect conceptual models for folding of the termini onto the rod. These models were generated using PyMol sculpting function.

- What does “Structural projection models” mean in Fig. 3? If they were generated manually, maybe replace by “Conceptual models” If they come from ensemble generated by Rosetta modeling, replace with “Example models” (as in Figure 2c).

As above, these are conceptual models. We have replaced all usages of “structural projection models” with “example models” throughout the text and legends.

- Extended data Fig. 5. Panels a-c are discussed together without a clear explanation of what is a, b, and c – hard to follow

We have modified the legend accordingly to explain these subpanels.

- Extended data Fig. 5d.: Line 520 speaks about “42 out of 48 input cross-links” while in the figure I can count only 14 crosslinks. The number of unique residue pairs should be used to calculate crosslink satisfaction numbers, otherwise the support looks artificially boosted (42 unique restraints is a lot!).

We agree that 42 out of 48 cross-links sounds artificially boosted. This was phrased like that following the actual input cross-links for Rosetta: the majority of cross-links around linker regions were possible as both intra-chain and inter-chain according to the calculated SILAC ratios. Therefore in search of models satisfying as many combinatorial combinations of cross-links as possible each cross-link was programmed as 2 (for intra-chain cross-links from chain A to chain A and chain B to chain B) or 4 (from any chain to any chain). In hindsight and after additional modelling experiments we note that this may have not been the best strategy (addressed further below) and consequently moved away from such phrasing to avoid contradiction between the text and the figures.

- The modeling of the starting fully stretched lamin A dimer based on homologous templates and with de novo fragments using CBuilder is reasonable at this resolution of modeling. However, what is the evidence that lamin A contains PH domain as vimentin, which was used for homology modeling of the PH region?

The evidence came from David Parry in (*Parry, D. A. Hendecad repeat in segment 2A and linker L2 of intermediate filament chains implies the possibility of a right-handed coiled-coil structure. J Struct Biol 155, 370-374, doi:10.1016/j.jsb.2006.03.017 (2006)*). To our knowledge no other evidence for a PH in lamin A has ever been produced. We note however that our own cross-linking data does not fit with the earlier prediction of the heptad substructure for this PH region and rather supports irregularity in its packing. This theoretically fits with the relatively low stability of the parallel hendecad folding in PH region, though we refrain from making any additional conclusions from limited data.

- In Methods, line 221: “Intra- and inter-molecular cross-links were used in duplicates (i.e. from chain A to chain A and from chain B to chain B)” shouldn’t inter-molecular cross-links be from chain A to B and B to A?

Thank you for noticing this mistake, this has been corrected.

- In Methods, lines 221-223 “Intra- and inter-molecular cross-links were used in duplicates (i.e. from chain A to chain A and from chain B to chain B), cross-links that happen both ways were used in tetraplets (A-A, A-B, B-A, B-B)” – this would artificially drive modeling to satisfy all possible pairs for a given crosslink while satisfaction of a single one is sufficient to explain the crosslink. In other words, this implementation implies a restraint: A-A AND A-B AND B-A AND B-B while it should be A-A OR A-B OR B-A OR B-B. The “AND” restraint may artificially lead to larger interfaces and more parallel conformations. Was the xWalk filtering also assuming “AND” restraint? This should be explained or modeling should be repeated or text should be changed to avoid proposing any specific conformation such as Z-folding

We very much thank the reviewer for this comment. This was not initially apparent in Rosetta’s algorithm but appeared to be true and reveals a hidden problem with cross-linking guided docking usage to model mutually exclusive interactions. As we discovered, when cross-links are used as constraints, Rosetta indeed models in an “AND” mode and attempts to satisfy the most cross-links often at the expense of otherwise favourable interfaces. Unfortunately there is no option to run guided docking in an “OR” mode in the current iteration and, considering the multiple possible combinatorial combinations it is not feasible to attempt multiple runs to satisfy them all. In an attempt to tackle this we therefore tried to run Rosetta docking for coils 1A and 1B adjacent termini fragments without constraints with the intention to then search for models with Xwalk that a) satisfy the constraints of linker backbone length and b) satisfy any of the cross-links. This resulted in a much larger set of models with stable (by I_sc) interfaces of all sorts:

even with the most stringent criteria in Xwalk we found over 1,500 parallel or close to parallel coil 1A/1B tandem stagger models that satisfied all 6 input in all possible variations. We note however that this eventually only serves to emphasise the potential redundancy in the overlap mechanism and does not add much to our results conceptually.

Regarding suggesting Z-folding as a specific conformation, the reviewer is correct that within the range of interaction scores obtained it will not likely be possible to pinpoint the most favourable conformation – in fact the “barrel-like” distribution of models shown for example in Supplementary Figure 3 for coil 1A/1B overlap suggests an extreme flexibility of the overlap interaction as whole. Thus we never aimed to suggest a single type of overlap based exclusively on the modelling data – only to demonstrate that overlap is possible and the interface is flexible. With this in mind we favour a parallel tandem stagger overlap because of the orthogonal input EM data – both our own and an extensive body of EM data from other labs before us on dimers and the tight packing of the assembled filaments. Correspondingly we attempted to emphasise this in the text as mentioned above.

- It would be beneficial to make the models publicly available as supplementary data or through dedicated databases such as <https://pdb-dev.wwpdb.org/>. Since multiple models are possible, they could be deposited as NMR-like model ensembles or localization densities (<https://www.ncbi.nlm.nih.gov/pubmed/29211988>).

By the time the manuscript if accepted is published these data will be publicly available on EdinburghShare.

REVIEWERS' COMMENTS:

Reviewer #1 (Remarks to the Author):

The authors have carried out both additional experiments and substantial additional analysis of their data and the points raised by the reviewer in the previous round of review have been satisfactorily addressed.

The manuscript is also in a more comprehensible format and easier to follow.

Once all the data including additional analyses, modelling runs etc are uploaded as announced to the Edinburgh Datashare and publicly available, this impressive work by Makarov et al. is ready for publication in Nature Communications.

And as a final side note - if the authors were indeed to decide to describe the structure as a set of Japanese nunchakus - this reviewer would not object.

Florian Stengel

Reviewer #2 (Remarks to the Author):

The authors have done a good job carefully considering the feedback from the three referees. Their detailed rebuttal letter is convincing and highlights a range of interesting points. The overall presentation has been very much improved by extending the main text and figures, greatly improving the readability of the manuscript.

In response to my initial judgement of the rotary shadowing experiments and analytical centrifugation data on dimers in solution as being less informative than the (very novel and interesting) cross-linking data, the authors insist that the former results are reliable and important. However, in the revised version these results are still presented very briefly (just 6 lines at the bottom of the first section of Results). I wonder if the authors could extend this section with more detail.

In particular, I would like the authors to include a discussion on the presence of higher oligomers in these experiments (Fig. 1). The conditions used here are the standard dimer conditions (pH8) rather than the assembly conditions (pH6). When presenting the UC data the authors only interpret them as indicating shorter dimers in phosphate. What about the tetramers, could a small fraction of tetramers have an effect here?

This is important to discuss since cross-linking in the dimer conditions nevertheless yields a distinct tetramer band. The resulting numerous cross-links are the basis of the whole modelling of the head-to-tail overlap (Fig. 5). How can the authors be sure that these cross-links are relevant, since the pH8 conditions are not supposed to induce assembly?

I did appreciate a careful revision of the ex vivo data, as these are quite crucial as already mentioned.

Reviewer #3 (Remarks to the Author):

In the revised version, Makarov et al. hold to their hypothesis that the length and flexibility of lamin A rod changes through electrostatically-driven folding of the coiled-coil segments onto each other around flexible linkers, but convey the message with much more clarity. From the new text and the responses, it is clear now that authors do not mean very specific conformations around the linkers, but a flexibility that allows the coiled-coil segments to overlap with transient electrostatic interfaces. To make it clearer, the authors renamed the Z-folding to tandem stagger, which is indeed less confusing. Most of my comments could be indeed resolved by simply clearer and more comprehensive descriptions. The modeling part has been extended for further analysis (modeling without crosslinks followed by filtering) and overall supports staggered conformations. In fact, since now it is clear that authors propose ensemble of models or example models out of those ensembles, the modeling merely strengthen the hypothesis anyway clearly visible from crosslinks and other data. The modeling adds that complementary surfaces do exist on the neighboring coiled coil segments and the linkers, and transient interactions of those could be responsible for the tandem stagger. In summary, the manuscript is interesting from both methodological (SILAC CLMS for studying homo-oligomers) and biological point (a plausible and now well supported mechanism for filament flexibility). It would be interesting in future to see an application of such methods (crosslinking integrated with rotary shadow EM and modeling) to other filaments, even those not formed by coiled-coils but globular domains connected with linkers (e.g. Ig-like domains).

RESPONSE TO REVIEWER'S COMMENTS

REVIEWERS' COMMENTS:

Reviewer #1 (Remarks to the Author):

The authors have carried out both additional experiments and substantial additional analysis of their data and the points raised by the reviewer in the previous round of review have been satisfactorily addressed.

The manuscript is also in a more comprehensible format and easier to follow.

Thank you, we appreciate the reviewer's comments as they have been particularly helpful in making this a better paper.

Once all the data including additional analyses, modelling runs etc are uploaded as announced to the Edinburgh Datashare and publicly available, this impressive work by Makarov et al. is ready for publication in Nature Communications.

This has now been done. Accession numbers for the mass spec data can be found at ProteomeXchange Consortium via the PRIDE ⁶⁸ partner repository with the dataset identifiers PXD008337 and PXD014009. Modelling data can be found on Edinburgh DataShare (<https://datashare.is.ed.ac.uk/handle/10283/3348>).

And as a final side note - if the authors were indeed to decide to describe the structure as a set of Japanese nunchakus - this reviewer would not object.

Florian Stengel

While we were very tempted to take the reviewer up on this offer, some lab members pointed out the potential for it to be viewed as a trigger for violence and so we have left it out.

Reviewer #2 (Remarks to the Author):

The authors have done a good job carefully considering the feedback from the three referees. Their detailed rebuttal letter is convincing and highlights a range of interesting points. The overall presentation has been very much improved by extending the main text and figures, greatly improving the readability of the manuscript.

We thank the reviewer for this positive assessment.

In response to my initial judgement of the rotary shadowing experiments and analytical centrifugation data on dimers in solution as being less informative than the (very novel and interesting) cross-linking data, the authors insist that the former results are reliable and important. However, in the revised version these results are still presented very briefly (just 6 lines at the bottom of the first section of Results). I wonder if the authors could extend this section with more detail.

In particular, I would like the authors to include a discussion on the presence of higher oligomers in these experiments (Fig. 1). The conditions used here are the standard dimer conditions (pH8) rather than the assembly conditions (pH6). When presenting the UC data the authors only interpret them as indicating

shorter dimers in phosphate. What about the tetramers, could a small fraction of tetramers have an effect here?

This is important to discuss since cross-linking in the dimer conditions nevertheless yields a distinct tetramer band. The resulting numerous cross-links are the basis of the whole modelling of the head-to-tail overlap (Fig. 5). How can the authors be sure that these cross-links are relevant, since the pH8 conditions are not supposed to induce assembly?

We thank the reviewer for highlighting both of these points. Regarding the detail, we have now expanded this section to give more detail about the analytical ultracentrifugation (AUC) experiments as shown in the additional text copied below. Regarding the issue of the presence of tetramers, indeed tetramers are present in the AUC data. However, they would not have an effect on the traces shown, but rather just appear at higher S-values between 5 and 6 S. In the initial version we cut this off so as to focus attention on the differences between the two buffers for the dimer traces; however, in this revision we have extended the traces from S=2.0-5.2 to S=2.0-6.0 so that the small tetramer peak can be visualised. We also show in the new supplementary figure 1 the AUC trace all the way out to 10 S so that it is clear that we either had extremely low abundance or non-detectable levels of higher order polymerisation. However, for tetramers, while much lower abundance than dimers, a clearly detectable AUC peak is observed between S values of 5 and 6 for the Tris buffer and the dimer peak in NaPi also sports a long trail with a shoulder that drops that drops between S values of 5 and 6. Thus, the cross-linking and AUC experiments match well in the dimeric and tetrameric species obtained where both were performed in Tris or NaPi buffers at pH 8.0. We have now elaborated this further in the main text by adding:

“...ultracentrifugation where an increase in more compact species with S-values higher than the 3.81 S calculated for a dimer with a 51 nm long rod was observed in NaPi buffer (Fig. 1d). These species are likely also dimeric because, while lamin A head-to-tail tetramers can be clearly observed by EM in either buffer, these are exceedingly rare (~ 1 per 20-25 dimers in NaPi) (Supplementary Figure 1a) and the 3.81 S peak accounts for the vast majority of material indicated all the way out to 10 S from this experiment (Supplementary Figure 1b). Furthermore, a separate smaller peak can be observed between 5 and 6 S

that fits the calculated predicted S-value for lamin head-to-tail tetramers of 5.0 S (Fig. 1d; Supplementary Figure 1b)."

With regards to the pH issue raised, as noted above the cross-linking and AUC experiments were both done in pH 8.0. We think that the reviewer's comment may indicate their thinking that there should be no tetrameric species at pH 8.0. However, in Ueli Aepli's experiments using rat liver purified lamin A he noted that pH 8.0 prolongs dimers by slowing further oligomerisation, but it does not prevent it. Moreover, note that in the Aepli 1986 Nature 323: 560-564 paper they performed sedimentation equilibrium experiments at twice the salt concentration we used (500 mM) and higher pH (pH 9.0) and noted "the lamins are a mixture of dimers and tetramers." Notably, in the *in vitro* assembly assays we dialysed the protein first from a pH 8.8 buffer (the pH generally used when trying to maintain the dimeric state) with urea into a pH 8.5 high salt buffer and then from the pH 8.5 buffer to a pH 6.5 buffer while at the same time dropping the salt concentration from 250 to 150 mM and moving from Tris to MES buffer. Thus, it is not surprising that, when dialysing for several hours into the pH 8.0 buffer, some oligomerisation would occur. And, as noted above, we indeed observe such oligomerisation in the form of tetramers at pH 8.0 by EM (Supplementary Figure 1).

I did appreciate a careful revision of the ex vivo data, as these are quite crucial as already mentioned.

We thank the reviewer for acknowledging the hard work the former student came back and did on this.

Reviewer #3 (Remarks to the Author):

In the revised version, Makarov et al. hold to their hypothesis that the length and flexibility of lamin A rod changes through electrostatically-driven folding of the coiled-coil segments onto each other around

flexible linkers, but convey the message with much more clarity. From the new text and the responses, it is clear now that authors do not mean very specific conformations around the linkers, but a flexibility that allows the coiled-coil segments to overlap with transient electrostatic interfaces. To make it clearer, the authors renamed the Z-folding to tandem stagger, which is indeed less confusing. Most of my comments could be indeed resolved by simply clearer and more comprehensive descriptions. The modeling part has been extended for further analysis (modeling without crosslinks followed by filtering) and overall supports staggered conformations. In fact, since now it is clear that authors propose ensemble of models or example models out of those ensembles, the modeling

merely strengthen the hypothesis anyway clearly visible from crosslinks and other data. The modeling adds that complementary surfaces do exist on the neighboring coiled coil segments and the linkers, and transient interactions of those could be responsible for the tandem stagger. In summary, the manuscript is interesting from both methodological (SILAC CLMS for studying homo-oligomers) and biological point (a plausible and now well supported mechanism for filament flexibility). It would be interesting in future to see an application of such methods (crosslinking integrated with rotary shadow EM and modeling) to other filaments, even those not formed by coiled-coils but globular domains connected with linkers (e.g. Ig-like domains).

We thank the reviewer both for their previous comments that helped us to make the manuscript more clear and for their strong support of the revised document. Indeed we are planning once this paper is out to submit a grant proposal to continue this work in which we will both compare this with other lamin subtypes and other IFs as well as apply the principles to design better polymers.